# *Divide and Conform*: Unleashing Spatial Filter Atoms for Unsupervised Target Transferability

## Abstract

The straightforward fine-tuning of the pre-trained model for the target task, bears the risk of under-utilizing the foundational knowledge accrued by the pre-trained model, resulting in the sub-optimal utilization of transferable knowledge, consequently impeding peak performance on the target task. To address this, we introduce *Divide and Conform*, aimed at augmenting the transferability of pre-trained convolutional neural networks (ConvNets), *in the absence of base data*. This strategy exploits the mathematical equivalence of the convolution operation, conceptualizing it as a two-step process involving spatial-only convolution and channel combination. To achieve this, we decompose (*Divide*) the filters of pre-trained ConvNets into spatial filter atoms (responsible for spatial-only convolution) and their corresponding atom-coefficients (responsible for channel combination). Our observations reveal that solely fine-tuning (*Conform*-ing) the spatial filter atoms, comprising of only a few hundred parameters, renders the transferability of the model efficient, without compromising on the predictive performance. Simultaneously, the static atom-coefficients serve to retain the base (foundational) knowledge from the pre-trained model. We rigorously assess this dual-faceted approach within the demanding and practical framework of cross-domain few-shot learning, showcasing the approach's substantial capability of transferring the knowledge in a parameter-efficient manner.

## 1 Introduction

In the wake of the transformative impact of language models (Devlin et al., 2019; Raffel et al., 2020; Touvron et al., 2023; Brown et al., 2020), the field of image processing is experiencing a parallel evolution. Language models, often pre-trained on internet text, contrast starkly with vision models pre-trained on internet images, which are typically not representative of specialized domains like medical or satellite imagery. This discrepancy in image types leads to under-performance when employing vision models pre-trained on generic images to specialized domains. Vision transformers (Dosovitskiy et al., 2021; Touvron et al., 2021) have marked a significant shift in vision models, yet recent findings (Smith et al., 2023; Liu et al., 2022; Woo et al., 2023) suggest that ConvNets can rival or even outperform transformers at scale. Despite this, the generalization capabilities of ConvNets, especially in the face of significant task discrepancies, remain a concern (Zhou et al., 2021; Bai et al., 2021).

Recent studies (Phoo & Hariharan, 2021; Oh et al., 2022; Li et al., 2021; Islam et al., 2021; Zhou et al., 2023; Fu et al., 2023; Li et al., 2022; Guo et al., 2020; Jiang et al., 2022) have highlighted a significant gap in the performance of pre-trained ConvNets when applied to target tasks that are markedly different, especially in situations where no labeled samples are available for the target task. In these scenarios, the effectiveness of pre-trained models falls short compared to their supervised counterparts. This observation underscores the necessity for further investigation into enhancing the transferability of pre-trained models, particularly in contexts where the domain gap is substantial and the availability of labeled data is non-trivial. This challenge, known as cross-domain few-shot learning (CD-FSL), prompted works by Phoo & Hariharan (2021), along with Islam et al. (2021), to utilize a small por-

tion of unlabeled samples from the target domain—merely a few thousand examples—to better align the pre-trained model with the target task, thereby enhancing its downstream utility.

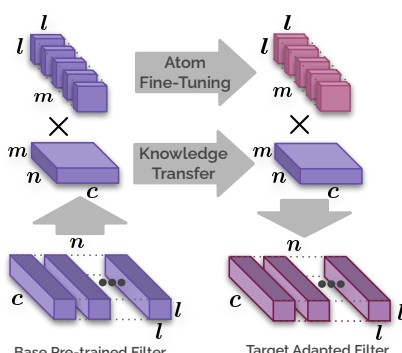

Figure 1: The figure illustrates the decomposition of pre-trained filters, initially trained on a base dataset, into two distinct components: *spatial atoms* and their associated *spatially-invariant atom-coefficients*. While the spatial atoms are subjected to selective fine-tuning, the atom-coefficients are preserved as-is. This approach effectively transfers the foundational knowledge, which is invariant to spatial discrepancies, to the target task at hand.

These studies (Phoo & Hariharan, 2021; Islam et al., 2021) emphasize the concurrent use of the large pre-training (base) and target data to either capitalize on or prevent the loss of essential, foundational knowledge contained within the pre-trained model. However, the practicality of simultaneous access to both datasets is often unrealistic in real-world settings, such as in a *client-vendor* setup (Kundu et al., 2021; 2020b;a), and it is prohibitively costly to reutilize the base dataset and the target dataset to fully recalibrate the model for the target task.

Therefore, in response to these identified challenges—**1)** the prevalent lack of the base dataset during target adaptation in practical applications, **2)** the parametric and computational cost associated with comprehensive fine-tuning, which can be prohibitively costly, and **3)** the scarcity of labeled target data, especially in domains necessitating expert annotation—we introduce our method, *Divide and Conform*. This approach is designed to enhance the adaptability of pre-trained models to specific target tasks, while assuming only a limited amount of unlabeled data is available for the target task and no access to the extensive base dataset, achieving all this in a parameter-efficient manner. Our methodology is structured around two principal components: the *Divide* phase and the *Conform* phase.

During the *Divide* phase, informed by the previous works of Zhang et al. (2015) and Qiu et al. (2018) showing that a convolution filter can be accurately represented as a linear combinations of a predefined set of spatial filter bases[1] (atoms), we exploit the inherent mathematical representation of the convolution operation as a sequence of *spatial-only convolution* and *channel combination*. This insight enables us to decompose the pre-trained filters into a set of spatial filter atoms (responsible for spatial-only convolution) and their associated atom-coefficients (responsible for channel combination). Note that the channel combination is conducted as a $1 \times 1$ convolution operation using the atom-coefficients, hence, the atom-coefficients could be said to capture the spatially-invariant knowledge.

Following this decomposition, the *Conform* phase aims to align the pre-trained knowledge towards the target task. The base pre-trained model is recalibrated by selectively fine-tuning the spatial atoms while keeping the atom-coefficients static. We hypothesize that the primary constituent of the variations between the base and the target task data can essentially be construed as *spatial* discrepancies in the images. Thus, by exclusively adapting the spatial atoms with a few unlabeled target samples, while keeping the atom-coefficients untouched, we seek to efficaciously recalibrate the pre-trained model for the target task, positing that this selective learning mechanism for the spatial atoms should suffice in aligning the pre-trained model to the target task's characteristics. This selective adaptation is conceptually similar to LoRA-style methods used in language models (Hu et al., 2021), where only a small, additionally targeted subset of parameters is fine-tuned while preserving the majority of the model's parameters. By focusing on adapting spatial atoms (akin to LoRA's low-rank updates), we reduce the number of trainable parameters, achieving parameter efficiency without sacrificing transferability. Moreover, this approach retains the spatially-invariant, parameter-expensive atom-coefficients, which capture the channel mixing (combination) knowledge inherited from the pre-trained model. In doing so, we ensure that the foundational knowledge from the base data is preserved for the target task, enhancing model transferability in constrained settings where comprehensive fine-tuning would be computationally prohibitive.

---

[1]We adopt the term *atoms* from dictionary learning literature to refer to subspace elements, noting that we do not impose orthogonality between them.

Finally, to demonstrate the efficiency and effectiveness of the transferability, we evaluate it in the challenging and practical CD-FSL setting (Guo et al., 2020; Phoo & Hariharan, 2021; Islam et al., 2021; Oh et al., 2022) due to its stringent evaluation protocol. In summary, the contributions of this work are as follows:

- We introduce a selective update approach that enhances the adaptability of a base pre-trained model to target tasks, without the necessity of accessing the base data.

- We align the pre-trained model towards target tasks by fine-tuning the decomposed spatial filter atoms using unlabeled target data samples, thereby addressing task discrepancies through spatial atom updates in an unsupervised manner.

- We aim to *retain* the essential channel-mixing knowledge, in the form of atom-coefficients, from the pre-trained filters, which aids in transferring foundational knowledge to the target task.

## 2 RELATED WORKS

**Cross-Domain Few-Shot Learning (CD-FSL).** CD-FSL addresses the challenge of transferring knowledge from a source to significantly different target domains (Guo et al., 2020; Tseng et al., 2020; Oh et al., 2022), where domain discrepancies hinder the direct application of learned knowledge (Song et al., 2023; Neyshabur et al., 2020). Recent progress emphasizes fine-tuning (tranfer-learning), outperforming traditional meta-learning based few-shot methods (Guo et al., 2020; Sung et al., 2018; Snell et al., 2017; Vinyals et al., 2016; Garcia & Bruna, 2018; Tseng et al., 2020; Sun et al., 2021; Wang & Deng, 2021; Hu & Ma, 2022). Notably, the prominent tranfer-learning based ideas, STARTUP (Phoo & Hariharan, 2021) and Dynamic Distillation (Islam et al., 2021) leverage small-scale unlabeled target data during pre-training to improve adaptability. Both methods train a teacher network on labeled base data, but STARTUP also trains a student network with unsupervised losses on target data, using distillation loss (Hinton et al., 2015) and SimCLR (Chen et al., 2020), while Dynamic Distillation employs a KL divergence loss (Sohn et al., 2020). TUP (Li et al., 2021) investigates pre-trained networks' clustering properties, recommending a few unlabeled target samples for better clustering. However, these methods necessitate access to the base data in addition to some unlabeled target samples during pre-training. Diverging from this, ConFeSS (Das et al., 2022) proposes a base-free adaptation strategy by learning a masking module that filters target-specific features, enhancing the model's relevance to the target task, however, they use labeled target data to align their base pre-trained model. Nonetheless, we use the CD-FSL setting as test-bed evaluating representation transferability (*cf*. Appendix A).

**Spatial Filter Decomposition.** Several studies have advanced the concept of approximating filters through parameter-efficient representations (Qiu et al., 2018; Li et al., 2019; Denton et al., 2014; Jaderberg et al., 2014; Zhang et al., 2015; Wang et al., 2017a). Among these, the contributions of Zhang et al. (2015) and Qiu et al. (2018) are particularly noteworthy for their proposal to represent filters as combinations of spatial bases (atoms). Building upon this decomposition framework, subsequent research has explored various applications, including, network compression (Li et al., 2019), domain adaptation (Wang et al., 2020), continual learning (Miao et al., 2021), and image generation (Wang et al., 2019; Zhai et al., 2021).

## 3 METHODOLOGY

### 3.1 PROBLEM SETTING & MOTIVATION

Our goal is to utilize a backbone $f$, pre-trained on a base dataset $D_B$ that is not accessible, to extract useful representations from target task data $D_T$. The significant disparity in sample characteristics between $D_B$ and $D_T$ can hinder the direct application of $f$, leading to inferior performance. To address this, we align the model with the target task using a subset of unlabeled target data $D_U$ ($D_U \subset D_T$), following (Phoo & Hariharan, 2021; Oh et al., 2022), to improve model adaptability in an unsupervised manner. For the purpose of evaluation the representation transferability, we adopt the CD-FSL framework (Oh et al., 2022), constructing episodic few-shot learning scenarios with support $D_S$ and query $D_Q$ sets from the labeled $D_L$ ($D_L \subset D_T$, $D_L \cap D_U = \varnothing$). Each episode

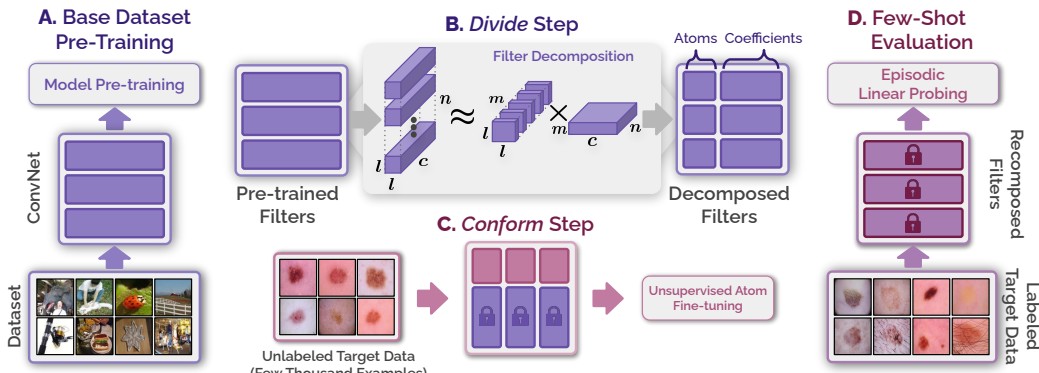

Figure 2: The figure illustrates the proposed methodology's workflow. **A.** depicts the model's pre-training on a large foundational dataset. In **B.**, the convolutional filters are decomposed into spatial filter atoms and atom-coefficients—our *Divide* step. **C.** shows that after decomposition, atom-coefficients are fixed while spatial atoms are fine-tuned with an unsupervised objective using unlabeled target task data—our *Conform* step. **D.** demonstrates the evaluation phase where the fine-tuned model, with the spatial atoms adjusted, is assessed in a few-shot scenario with labeled target data, after freezing the backbone and attaching a linear classifier.

features $c$ classes and selects $k$ instances per class for $D_S$ and $k_q$ instances (usually 15) for $D_Q$, ensuring no overlap. We train a linear classifier $g$ on $D_S$ using features from frozen $f$, and test $g \circ f$ on $D_Q$. Our setup uses $c = 5$ and $k \in \{1, 5\}$, averaging accuracy over 600 episodes as per previous studies (Guo et al., 2020; Phoo & Hariharan, 2021; Oh et al., 2022).

A critical consideration in this process is the potential risk of naively fine-tuning the entire network $f$ with $D_U$, which could inadvertently overwrite the model's intrinsic, foundational knowledge, acquired from the now inaccessible $D_B$. Such an approach not only risks diluting the pre-learned knowledge but also incurs a high parameter update cost. To mitigate these concerns, we propose a strategy termed *Divide and Conform*. The initial phase, *Divide*, entails the decomposition of convolution filters into spatial filter atoms and their respective atom-coefficients (Qiu et al., 2018; Li et al., 2019; Zhang et al., 2015). Subsequently, the *Conform* phase involves the selective fine-tuning of these spatial atoms, whilst maintaining the atom-coefficients unchanged, using unsupervised learning objectives (Chen et al., 2020; Grill et al., 2020). The key idea behind keeping the spatially-invariant atom-coefficients static is to *retain* the foundational knowledge embedded within $f$, and advance the model's adaptation in a parameter-efficient manner, as the fine-tuning parameters (spatial filter atoms) account for only a few hundred parameters. This nuanced approach ensures the retention of $f$'s foundational strengths while facilitating its tailored application to samples in $D_T$, thereby striking a favorable balance between performance and efficiency. In Figure 2, we visually describe the overall pipeline of the problem setup and our methodology. In the upcoming sections we discuss the *Divide* and the *Conform* steps in detail.

### 3.2 THE *Divide* STEP

In a typcial convolution, with a single stride. For the input feature map $\mathbf{I} \in \mathbb{R}^{c \times h \times w}$, with dimensions height $h$, width $w$, and channels $c$, and a convolutional kernel $\mathbf{K} \in \mathbb{R}^{n \times c \times l \times l}$ where $l$ is the kernel size and $n$ the number of kernels, the convolution output $\mathbf{O} \in \mathbb{R}^{n \times h \times w}$ is obtained by applying $\mathbf{K}$ to $\mathbf{I}$ (with suitable padding to preserve spatial dimensions). The operation is defined as:

$$\mathbf{O}[j, y, x] = \sum_{i=1}^{c} \sum_{u=-\delta}^{\delta} \sum_{v=-\delta}^{\delta} \mathbf{K}[j, i, u, v] \cdot \mathbf{I}[i, y - u, x - v], \tag{1}$$

where $\delta = \lfloor l/2 \rfloor$, ensuring $1 \leq y \leq h$, $1 \leq x \leq w$, and $1 \leq j \leq n$.

Zhang et al. (2015) initially posited, and subsequently Qiu et al. (2018) and Li et al. (2019) further substantiated, the concept that a convolutional filter can be accurately represented as a linear combination of a pre-defined set of spatial filter bases. Building upon this foundational work, we

incorporate a similar methodology for filter decomposition, as depicted in Figure 2B. Here, a convolutional filter is represented as a linear combination of $m$ spatial atoms $\mathbf{D} \in \mathbb{R}^{m \times l \times l}$, where the relationship $\mathbf{K} = \mathbf{AD}$ is established, and $\mathbf{A} \in \mathbb{R}^{n \times c \times m}$ denotes the coefficients of composition. This decomposition allows the convolution operation to be reinterpreted as a two-step process. The initial step involves executing spatial convolutions individually with each of the $m$ filter atoms, delineated as follows:

$$\mathbf{O}'[k,i,y,x] = \underbrace{\sum_{u=-\delta}^{\delta} \sum_{v=-\delta}^{\delta} \mathbf{D}[k,u,v] \cdot \mathbf{I}[i, y-u, x-v]}_{\text{Spatial-only Atom Convolution}}. \tag{2}$$

$\mathbf{O}' \in \mathbb{R}^{m \times c \times l \times l}$ signifies the interim output post the convolution with $\mathbf{D}$, for $1 \leq k \leq m$. Then, the final output $\mathbf{O}$ is obtained by linearly combining the intermediate features of $\mathbf{O}'$ using $\mathbf{A}$, as illustrated below:

$$\mathbf{O}[j,y,x] = \underbrace{\sum_{i=1}^{c} \sum_{k=1}^{m} \mathbf{A}[j,i,k] \cdot \mathbf{O}'[k,i,y,x]}_{\text{Channel Combination}}. \tag{3}$$

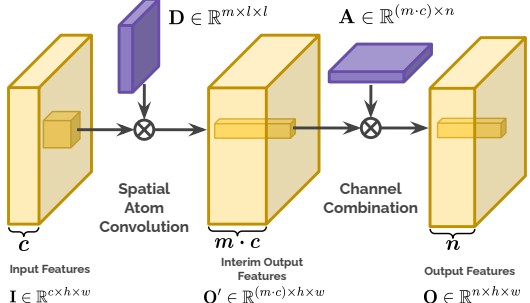

Figure 3: The figure depicts a convolution operation executed in two phases, utilizing spatial atoms ($\mathbf{D}$) and their associated atom-coefficients ($\mathbf{A}$). This sequential convolution is functionally equivalent to a direct convolution using the composite filter ($\mathbf{K} = \mathbf{AD}$).

Practically, this subsequent step is executed through a $1 \times 1$ convolution, leveraging $\mathbf{A}$ as a spatially invariant linear transformation. Moreover, given the linear nature of all involved operations, this two-step approach precisely equals the effect of convolving with the fully reconstructed filter. The two-step convolution described in (2) and (3) is demonstrated in Figure 3.

Hence, to decompose the dense, pre-trained filters, we formulate the filter decomposition objective as a dictionary learning problem (Mairal et al., 2009; Chen et al., 1998), minimizing the least-squares objective. The involved tensors are reshaped accordingly: $\mathbf{K} \xrightarrow{\text{reshape}} \mathbf{K} \in \mathbb{R}^{(n \cdot c) \times l^2}$, $\mathbf{D} \xrightarrow{\text{reshape}} \mathbf{D} \in \mathbb{R}^{m \times l^2}$, and $\mathbf{A} \xrightarrow{\text{reshape}} \mathbf{A} \in \mathbb{R}^{(n \cdot c) \times m}$. Furthermore, to prevent $\mathbf{D}$ from becoming arbitrarily large, which would result in correspondingly small values of $\mathbf{A}$, as a customary step (Mairal et al., 2009), we constrain the columns of $\mathbf{D}$ to have an $\ell_2$ norm of at most one. The convex set of matrices satisfying this constraint, denoted by $\mathcal{C}$, is defined as:

$$\mathcal{C} \triangleq \{\mathbf{D} \in \mathbb{R}^{m \times l^2}) \mid \text{s.t. } \text{diag}(\mathbf{D}^{\top}\mathbf{D}) \leq \mathbf{1}\}. \tag{4}$$

Finally, the objective function for decomposing $\mathbf{K}$, with a sparsity regularization on $\mathbf{A}$, is formulated as:

$$\min_{\mathbf{D} \in \mathcal{C}, \mathbf{A} \in \mathbb{R}^{(n \cdot c) \times m}} \frac{1}{n \cdot c} \sum_{i=1}^{n \cdot c} \left( \frac{1}{2} \|\mathbf{K}[i] - \mathbf{A}[i] \cdot \mathbf{D}\|_2^2 + \lambda \|\mathbf{A}[i]\|_1 \right). \tag{5}$$

The above objective is solved by altering the optimization between the two variables ($\mathbf{D}$ and $\mathbf{A}$), minimizing over one while keeping the other one fixed, as proposed by Lee et al. (2006). Note that we adopt an iterative strategy as opposed to a closed-form solution to allow for overparametrization *i.e.* $m \geq l^2$ atoms. In Appendix Table 13, we analyze the decomposition residual error that we observed for different number of filter atoms, and observe a notable reduction in the decomposition error for filter atoms 9 and beyond and therefore we limit our analysis to 9 and 12 atoms. In Appendix E.3 we analyze the downstream performance for different $\lambda$ values.

Table 1: **5-way 1-shot and 5-shot** accuracy (%) with 95% confidence using **ResNet-18 and ImageNet-1K**, detailing unlabeled data use (first column) and parameters fine-tuned (last column). Best and second-best results are highlighted in **maroon** and **navy**. Our method outperforms **Sim-CLR** Chen et al. (2020) and **LoRA** Hu et al. (2021) on majority of the datasets, while fine-tuning less than **2%** of the total number parameters in the vanilla backbone.

| % Unlabeled Data | Method | Dataset | | | | | | | | # of Params. |
| | | ChestX | ISIC | EuroSAT | Crop | CUB | Cars | Places | Plantae | |
| 5-way 1-shot | | | | | | | | | | |
| 1% | SimCLR (Chen et al., 2020) | 22.20±0.41 | 32.19±0.58 | **70.76±0.90** | 82.56±0.86 | 44.22±0.85 | 32.71±0.67 | **59.85±0.91** | 41.48±0.82 | 11,176.51K |
| | LoRA-3 (Hu et al., 2021) | 22.38±0.42 | 31.65±0.54 | **69.96±0.89** | **83.36±0.84** | 44.41±0.85 | 32.60±0.65 | 59.07±0.92 | 40.71±0.78 | 218.75K |
| | DC-9 (Ours) | **22.87±0.48** | **33.73±0.59** | 69.10±0.85 | 82.35±0.79 | **50.61±0.88** | **34.42±0.67** | 59.50±0.86 | **42.92±0.77** | 192.34K (98.28%↓) |
| | DC-12 (Ours) | **22.62±0.42** | **34.34±0.59** | 68.88±0.83 | **83.28±0.81** | 49.19±0.86 | 33.61±0.66 | 59.98±0.88 | **43.31±0.81** | 192.77K (98.28%↓) |
| 5% | SimCLR (Chen et al., 2020) | 21.86±0.41 | 34.59±0.60 | **81.92±0.77** | 89.66±0.76 | 42.13±0.86 | 35.11±0.73 | 64.76±0.91 | 45.20±0.85 | 11,176.51K |
| | LoRA-3 (Hu et al., 2021) | 21.86±0.41 | 35.20±0.64 | **81.90±0.76** | **91.68±0.69** | 41.84±0.84 | 34.38±0.70 | **65.26±0.92** | 45.16±0.84 | 218.75K |
| | DC-9 (Ours) | **22.45±0.42** | **37.20±0.65** | 77.24±0.80 | 88.47±0.70 | **46.34±0.84** | **35.91±0.70** | 64.07±0.86 | **46.29±0.82** | 192.34K (98.28%↓) |
| | DC-12 (Ours) | **22.48±0.42** | **35.73±0.61** | 77.88±0.76 | 88.60±0.70 | **49.48±0.86** | **36.06±0.70** | 64.27±0.84 | 46.02±0.82 | 192.77K (98.28%↓) |
| 5-way 5-shot | | | | | | | | | | |
| 1% | SimCLR (Chen et al., 2020) | 25.01±0.42 | 42.14±0.55 | 84.62±0.54 | 92.96±0.48 | 63.27±0.81 | 47.50±0.76 | 76.34±0.62 | 58.19±0.79 | 11,176.51K |
| | LoRA-3 (Hu et al., 2021) | 24.79±0.43 | 41.93±0.57 | 84.76±0.54 | 93.33±0.46 | 63.13±0.81 | 47.46±0.75 | 75.92±0.64 | 57.27±0.77 | 218.75K |
| | DC-9 (Ours) | **26.63±0.43** | **45.65±0.60** | **86.65±0.48** | **94.42±0.39** | **73.74±0.76** | **52.28±0.75** | **80.09±0.58** | **61.23±0.76** | 192.34K (98.28%↓) |
| | DC-12 (Ours) | **26.47±0.46** | **46.35±0.57** | **86.34±0.49** | **94.64±0.41** | 71.89±0.78 | 51.25±0.75 | **80.46±0.58** | **61.92±0.75** | 192.77K (98.28%↓) |
| 5% | SimCLR (Chen et al., 2020) | 25.01±0.42 | 46.71±0.47 | **92.51±0.38** | 96.72±0.36 | 58.48±0.81 | 48.60±0.81 | 80.17±0.58 | 61.90±0.76 | 11,176.51K |
| | LoRA-3 (Hu et al., 2021) | 24.92±0.42 | 46.15±0.56 | **92.36±0.37** | **97.37±0.31** | 58.99±0.80 | 48.22±0.80 | 80.40±0.57 | 61.64±0.78 | 218.75K |
| | DC-9 (Ours) | **26.70±0.45** | **49.96±0.57** | 91.75±0.38 | 97.32±0.29 | **70.25±0.76** | **54.21±0.78** | 81.87±0.55 | **64.63±0.80** | 192.34K (98.28%↓) |
| | DC-12 (Ours) | **26.67±0.44** | **48.95±0.57** | 91.92±0.38 | **97.37±0.28** | 70.12±0.76 | 53.71±0.79 | **82.10±0.56** | **64.62±0.79** | 192.77K (98.28%↓) |

## 3.3 THE *Conform* STEP

The *Conform* step is executed to align the pre-trained backbone $f$ with the target task by leveraging unlabeled target samples. This alignment is informed by seminal contributions from the self-supervised learning (SSL) literature (Chen et al., 2020; He et al., 2020; Chen & He, 2021; Grill et al., 2020). In particular, we utilize the SimCLR (Chen et al., 2020) framework that employs $D_U$ for fine-tuning $f$, as the baseline, *cf*. Appendix B for further details on the finetuning objective.

Table 2: Parameter count comparison for ResNet-50 variants.

| DC Variant | # of Params. |
| --- | --- |
| Vanilla | 23,508.03K |
| **LoDC-9** | 437.07K (**98.14%↓**) |
| **LoDC-12** | 561.92K (**97.61%↓**) |
| **DC-9** | 12,192.08K (**48.14%↓**) |
| **DC-12** | 12,192.51K (**48.13%↓**) |

Moving forward, the observations from (2), (3), and Figure 3 demonstrate that, within the discussed filter decomposition framework, the convolution operation is disentangled into spatial-only atom convolution and channel combination operations. Based on this observation, we argue that exclusively fine-tuning the atoms with the unlabeled target data should suffice in aligning the model to the target task. This hypothesis is supported empirically (*cf*. Section 4.1) and by the following analysis:

Consider $\mathbf{K}_B$ and $\mathbf{K}_T$ as the convolutional filters for the base and target, respectively, decomposed under common composition coefficients $\mathbf{A}$. These filters can be expressed as:

$$\mathbf{K}_B(\mathbf{I}) = \mathbf{A} \cdot \mathbf{D}_B(\mathbf{I}), \quad \mathbf{K}_T(\mathbf{I}) = \mathbf{A} \cdot \mathbf{D}_T(\mathbf{I}). \tag{6}$$

It can be demonstrated that transformations on the entire filter can be achieved by manipulating the spatial atoms:

1. ***Insight 1***: Consider a linear transformation $\Theta : \mathbb{R} \mapsto \mathbb{R}$ that captures the atom transformation $\mathbf{D}_B \mapsto \mathbf{D}_T = \Theta(\mathbf{D}_B)$, then it can be shown that $\mathbf{K}_B \mapsto \mathbf{K}_T = \Theta(\mathbf{K}_B)$:

$$\Theta(\mathbf{K}_B) \Leftrightarrow \Theta(\mathbf{A} \cdot \mathbf{D}_B) \Leftrightarrow \mathbf{A} \cdot \Theta(\mathbf{D}_B) \Leftrightarrow \mathbf{A} \cdot \mathbf{D}_T \Leftrightarrow \mathbf{K}_T. \tag{7}$$

2. ***Insight 2***: For a given spatial transformation $\pi$, say translation, characterized by $\mathbf{\Gamma}_o^\pi(\mathbf{K}(\mathbf{I})) = \mathbf{K}(\mathbf{\Gamma}_i^\pi(\mathbf{I}))$, where $\mathbf{\Gamma}_i^\pi$ and $\mathbf{\Gamma}_o^\pi$ denote $\pi$'s representation in the input and output (post-convolution with $\mathbf{K}$) space, respectively, ensuring equality. Then, it can be inferred that the transformation's effect on the output of the filter is mediated through the spatial atoms. Hence, if there exists $\mathbf{D}_B(\mathbf{I}) \mapsto \mathbf{D}_T(\mathbf{I}) = \mathbf{\Gamma}_o^\pi(\mathbf{D}_B(\mathbf{I}))$ then it can be shown that $\mathbf{K}_B(\mathbf{I}) \mapsto \mathbf{K}_T(\mathbf{I}) = \mathbf{\Gamma}_o^\pi(\mathbf{K}_B(\mathbf{I}))$, akin to (7).

Hence, In the *Conform* step we only update the spatial atoms ($\mathbf{D}$) using the SimCLR objective (delineated in (8) and (9) in the Appendix), while keeping the atom-coefficients ($\mathbf{A}$) static.

Furthermore, the insights developed above can, in principle, be generalized to linear layers and, by extension, applied to architectures such as the Vision Transformer (ViT) (Dosovitskiy et al., 2021).

Table 3: **5-way 1-shot and 5-shot** accuracy (%) with 95% confidence using **ResNet-50 and ImageNet-1K**, detailing unlabeled data use (first column) and parameters fine-tuned (last column). Best and second-best results are highlighted in **maroon** and **navy**. Again, our method outperforms **SimCLR** (Chen et al., 2020) and **LoRA** (Hu et al., 2021) on majority of the datasets, while fine-tuning less than **2%-4%** of the total number parameters in the vanilla backbone.

| % Unlabeled Data | Method | Dataset | | | | | | | | # of Params. |
|---|---|---|---|---|---|---|---|---|---|---|
| | | ChestX | ISIC | EuroSAT | Crop | CUB | Cars | Places | Plantae | |
| | | | | | **5-way 1-shot** | | | | | |
| 1% | SimCLR (Chen et al., 2020) | 22.14±0.40 | 30.47±0.52 | 72.56±0.88 | 84.32±0.86 | 39.59±0.80 | 30.49±0.64 | 58.22±0.93 | 43.21±0.84 | 23,508.03K |
| | LoRA-4 (Hu et al., 2021) | 22.41±0.42 | 31.17±0.57 | 71.49±0.86 | 84.66±0.83 | 45.06±0.86 | 28.77±0.61 | 60.03±0.94 | 42.10±0.81 | 504.01K |
| | LoDC-9 (Ours) | 22.56±0.44 | 34.42±0.60 | 71.61±0.83 | 84.04±0.80 | 52.44±0.85 | 32.90±0.66 | 63.54±0.88 | 44.39±0.81 | 437.07K (98.14%↓) |
| | LoDC-12 (Ours) | 22.61±0.43 | 33.40±0.59 | 71.64±0.81 | 84.09±0.81 | 53.33±0.87 | 34.97±0.68 | 63.67±0.88 | 44.28±0.80 | 561.92K (97.61%↓) |
| 5% | SimCLR (Chen et al., 2020) | 21.81±0.41 | 35.62±0.65 | 85.53±0.74 | 91.59±0.69 | 41.52±0.85 | 34.55±0.72 | 68.01±0.91 | 46.16±0.88 | 23,508.03K |
| | LoRA-4 (Hu et al., 2021) | 21.91±0.40 | 34.42±0.60 | 83.61±0.73 | 91.33±0.71 | 40.63±0.83 | 35.61±0.73 | 67.70±0.92 | 47.21±0.88 | 504.01K |
| | LoDC-9 (Ours) | 21.95±0.42 | 35.05±0.61 | 79.57±0.73 | 88.84±0.71 | 50.48±0.89 | 35.20±0.70 | 68.06±0.84 | 48.13±0.85 | 437.07K (98.14%↓) |
| | LoDC-12 (Ours) | 22.22±0.43 | 35.37±0.61 | 80.53±0.76 | 88.09±0.71 | 51.14±0.90 | 34.84±0.73 | 67.78±0.82 | 48.07±0.86 | 561.92K (97.61%↓) |
| | | | | | **5-way 5-shot** | | | | | |
| 1% | SimCLR (Chen et al., 2020) | 24.55±0.41 | 40.17±0.54 | 86.04±0.51 | 93.64±0.46 | 54.95±0.81 | 42.74±0.71 | 75.56±0.63 | 60.13±0.79 | 23,508.03K |
| | LoRA-4 (Hu et al., 2021) | 24.86±0.43 | 40.97±0.59 | 85.96±0.48 | 93.78±0.44 | 63.36±0.79 | 38.62±0.68 | 76.56±0.65 | 59.18±0.77 | 504.01K |
| | LoDC-9 (Ours) | 25.60±0.43 | 46.40±0.57 | 87.46±0.48 | 94.62±0.42 | 74.72±0.77 | 48.16±0.76 | 82.02±0.57 | 63.00±0.77 | 437.07K (98.14%↓) |
| | LoDC-12 (Ours) | 25.47±0.44 | 45.63±0.57 | 87.63±0.47 | 94.47±0.42 | 75.98±0.76 | 52.36±0.77 | 82.20±0.58 | 62.95±0.78 | 561.92K (97.61%↓) |
| 5% | SimCLR (Chen et al., 2020) | 25.04±0.42 | 46.50±0.59 | 93.04±0.36 | 97.59±0.31 | 57.65±0.81 | 47.04±0.77 | 82.38±0.55 | 63.12±0.80 | 23,508.03K |
| | LoRA-4 (Hu et al., 2021) | 24.94±0.42 | 46.20±0.59 | 93.18±0.35 | 97.34±0.32 | 55.76±0.82 | 48.83±0.79 | 81.83±0.55 | 64.23±0.79 | 504.01K |
| | LoDC-9 (Ours) | 25.68±0.44 | 48.00±0.58 | 92.34±0.37 | 97.31±0.29 | 71.20±0.78 | 50.69±0.76 | 85.05±0.51 | 67.00±0.77 | 437.07K (98.14%↓) |
| | LoDC-12 (Ours) | 25.89±0.42 | 48.57±0.60 | 92.72±0.37 | 97.08±0.30 | 71.98±0.76 | 50.43±0.78 | 85.03±0.50 | 66.78±0.79 | 561.92K (97.61%↓) |

Specifically, one could decompose the weight matrices of linear layers into analogous components, potentially enabling similar manipulations. However, unlike ConvNets, where convolutional filters possess an inherent spatial structure that allows for a clear separation between spatial atoms and channel combination coefficients, linear layers lack such intrinsic spatial interpretability. The decomposed components in linear layers do not correspond to distinct functional units related to spatial or channel-wise operations, rendering the interpretability of such decomposition less evident.

Moreover, when decomposing convolutional filters into spatial atoms and atom-coefficients within ConvNets, we observe a pronounced imbalance in parameter distribution: the spatial atoms constitute a significantly smaller subset of the total parameters compared to the atom-coefficients. This imbalance is advantageous for our *Conform* step, as it permits fine-tuning of only the spatial atoms—a relatively small number of parameters, *cf*. Section 3.4.

### 3.4 PARAMETER EFFICIENCY ANALYSIS

Furthermore, decomposing convolutional filters into spatial atoms $\mathbf{D} \in \mathbb{R}^{m \times l \times l}$ and atom-coefficients $\mathbf{A} \in \mathbb{R}^{n \times c \times m}$ reveals a significant disparity in parameter counts between these components. The spatial atoms consist of $ml^2$ parameters, while the atom-coefficients comprise $mnc$ parameters. Typically, the product $nc$ vastly exceeds $l^2$ (*i.e.*, $nc \gg l^2$), indicating that the spatial atoms represent a relatively small fraction of the total parameters in the filter decomposition. To illustrate this imbalance, consider a convolutional layer with filters of dimensions $64 \times 64 \times 3 \times 3$, corresponding to 64 output channels, 64 input channels, and a kernel size of $3 \times 3$. The total number of parameters in this layer is $64 \times 64 \times 3 \times 3 = 36,864$. When decomposed into $m = 9$ spatial atoms of size $3 \times 3$, the spatial atoms account for only $9 \times 3 \times 3 = 81$ parameters. In contrast, the atom-coefficients retain $64 \times 64 \times 9 = 36,864$ parameters, matching the original filter's parameter count. This pronounced parameter imbalance is strategically leveraged in our *Conform* step. By fine-tuning only the spatial atoms $\mathbf{D}$-while keeping the atom-coefficients $\mathbf{A}$ fixed—we drastically reduce the number of parameters that need updating during adaptation to the target task. Specifically, we adjust a mere 81 parameters instead of the full $36,864$ parameters required when fine-tuning the entire filter. This selective fine-tuning not only enhances computational efficiency but also mitigates the risk of overfitting, as fewer parameters are susceptible to noise from the unlabeled data.

## 4 EXPERIMENTS AND ANALYSIS

### 4.1 RESULTS AND OBSERVATIONS

**Analysis with Few Unlabeled Samples.** In Table 1, we conduct a comparative analysis of the traditional fine-tuning method using the SimCLR objective, which updates all parameters, against *Divide and Conform* (DC), on the ResNet-18 backbone. We detail the performance of two DC variants, DC-9 and DC-12, which utilize 9 and 12 spatial filter atoms, respectively. Our method stands

Table 4: **5-way 1-shot and 5-shot** accuracy (%) with 95% confidence using **ResNet-18 and ImageNet-1K** in the standard CD-FSL setting (Oh et al., 2022; Phoo & Hariharan, 2021). The second column (**BF**-Base Free) indicates the absence of base data during target fine-tuning. The last column indicates the number of parameters fine-tuned. Best and second-best results, among **BF** methods, are highlighted in **maroon** and **navy**. Our method outperforms the **BF** methods on majority of the datasets, while fine-tuning less than **1-2%** of the total number parameters in the vanilla backbone.

| Method | BF | ChestX | ISIC | EuroSAT | Crop | CUB | Cars | Places | Plantae | # of Params. |
|---|---|---|---|---|---|---|---|---|---|---|
| | | | | | 5-way 1-shot | | | | | |
| STARTUP (Phoo & Hariharan, 2021) | ✗ | 23.03±0.42 | 31.69±0.59 | 73.83±0.77 | 85.10±0.74 | 72.58±0.93 | 45.75±0.84 | 66.02±0.87 | 49.78±0.93 | 11,176.51K |
| DynDistill (Islam et al., 2021) | ✗ | 24.02±1.59 | 34.55±1.82 | 77.24±1.06 | 87.53±1.01 | 63.80±1.32 | 46.55±1.21 | 60.84±1.08 | 49.90±1.22 | 11,176.51K |
| SimCLR (Chen et al., 2020) | ✓ | 21.52±0.41 | 35.35±0.67 | 85.59±0.71 | 92.16±0.74 | 41.77±0.87 | 35.98±0.76 | 70.08±0.94 | 48.48±0.93 | 11,176.51K |
| LoRA-12 (Hu et al., 2021) | ✓ | 21.58±0.42 | 35.29±0.68 | 86.43±0.70 | 91.85±0.76 | 42.03±0.89 | 36.13±0.77 | 69.69±094 | 48.42±0.90 | 846.18K |
| LoRA-9 (Hu et al., 2021) | ✓ | 21.64±0.42 | 35.26±0.68 | 85.71±0.71 | 92.20±0.76 | 41.95±0.87 | 36.43±0.79 | 70.17±0.93 | 48.46±0.90 | 637.03K |
| LoRA-3 (Hu et al., 2021) | ✓ | 21.62±0.41 | 35.27±0.66 | 85.80±0.70 | 92.11±0.75 | 41.81±0.87 | 36.65±0.78 | 69.89±0.94 | 48.68±0.91 | 218.75K |
| **LoDC-9 (Ours)** | ✓ | **22.36±0.42** | 33.61±0.61 | 77.64±0.78 | 85.37±0.74 | 48.07±0.86 | 35.98±0.68 | 65.47±0.89 | 46.41±0.81 | **32.40K (99.71%↓)** |
| **LoDC-12 (Ours)** | ✓ | **22.47±0.41** | 33.95±0.60 | 77.25±0.79 | 85.41±0.74 | 48.38±0.86 | 36.28±0.71 | 66.34±0.88 | 46.75±0.83 | **36.86K (99.67%↓)** |
| **DC-9 (Ours)** | ✓ | 22.16±0.42 | 36.03±0.65 | 81.35±0.73 | 89.98±0.69 | 47.89±0.85 | 36.99±0.72 | 66.61±0.91 | 49.22±0.86 | **192.34K (98.28%↓)** |
| **DC-12 (Ours)** | ✓ | 22.12±0.42 | 36.48±0.65 | 81.42±0.74 | 89.72±0.69 | 48.05±0.86 | 36.86±0.75 | 66.90±0.90 | 49.02±0.87 | **192.77K (98.28%↓)** |
| | | | | | 5-way 5-shot | | | | | |
| STARTUP (Phoo & Hariharan, 2021) | ✗ | 27.40±0.46 | 46.02±0.59 | 89.70±0.41 | 96.06±0.33 | 89.60±0.55 | 68.43±0.82 | 85.00±0.52 | 69.40±0.84 | 11,176.51K |
| DynDistill (Islam et al., 2021) | ✗ | 29.65±0.67 | 50.06±0.86 | 92.28±0.46 | 97.60±0.35 | 86.54±1.88 | 69.45±1.12 | 82.22±0.81 | 71.49±1.06 | 11,176.51K |
| SimCLR (Chen et al., 2020) | ✓ | 24.45±0.44 | 47.61±0.61 | 95.11±0.28 | 97.54±0.37 | 55.54±0.87 | 48.32±0.84 | 84.40±0.53 | 65.85±0.83 | 11,176.51K |
| LoRA-12 (Hu et al., 2021) | ✓ | 24.43±0.41 | 47.75±0.61 | 95.46±0.26 | 97.47±0.37 | 56.06±0.88 | 48.94±0.83 | 84.56±0.54 | 65.77±0.83 | 846.18K |
| LoRA-9 (Hu et al., 2021) | ✓ | 24.51±0.42 | 47.32±0.57 | 95.25±0.27 | 97.47±0.35 | 55.94±0.87 | 48.87±0.84 | 84.85±0.50 | 65.57±0.85 | 637.03K |
| LoRA-3 (Hu et al., 2021) | ✓ | 24.46±0.43 | 47.74±0.60 | 95.33±0.27 | 97.45±0.35 | 56.06±0.87 | 48.82±0.83 | 84.48±0.54 | 65.90±0.83 | 218.75K |
| **LoDC-9 (Ours)** | ✓ | **26.65±0.44** | 46.20±0.60 | 91.55±0.39 | 96.54±0.33 | 70.58±0.78 | 55.10±0.77 | 82.76±0.55 | 64.36±0.79 | **32.40K (99.71%↓)** |
| **LoDC-12 (Ours)** | ✓ | **26.60±0.45** | 46.82±0.60 | 91.54±0.38 | 96.49±0.34 | 70.68±0.76 | 54.96±0.76 | 82.31±0.56 | 64.63±0.81 | **36.86K (99.67%↓)** |
| **DC-9 (Ours)** | ✓ | 26.50±0.45 | 49.02±0.62 | 93.81±0.29 | 97.60±0.30 | 70.05±0.80 | 54.04±0.60 | 83.44±0.55 | 67.74±0.79 | **192.34K (98.28%↓)** |
| **DC-12 (Ours)** | ✓ | 26.40±0.46 | 50.14±0.64 | 93.90±0.31 | 97.61±0.30 | 70.65±0.78 | 55.34±0.78 | 83.60±0.55 | 67.41±0.81 | **192.77K (98.28%↓)** |

Table 5: **5-way 1-shot and 5-shot** CD-FSL performance (%) with 95% confidence intervals using **ResNet-10 with *mini*ImageNet** in the standard setting (Oh et al., 2022; Phoo & Hariharan, 2021). The last column indicates the number of parameters fine-tuned. The best and the second-best results are highlighted in **maroon** and **navy**, respectively. For Transfer-Learning-based methods, ✗ denotes access to base dataset and * denotes access to labeled target data. Parameter counts for Meta-Learning-based methods are not provided due to their varied learning strategies; however, all reported results utilize the ResNet-10 backbone.

| | Method | ChestX | ISIC | EuroSAT | Crop | CUB | Cars | Places | Plantae | # of Params. |
|---|---|---|---|---|---|---|---|---|---|---|
| | | | | | 5-way-1-shot | | | | | |
| Meta-Learning | FT (Guo et al., 2020) | 22.88±0.42 | 29.91±0.54 | 65.03±0.88 | 72.82±0.87 | 40.56±0.78 | 30.20±0.54 | 52.45±0.78 | 36.72±0.67 | - |
| Meta-Learning | RelationNet (Sung et al., 2018) | - | - | - | - | 42.44±0.77 | 29.11±0.60 | 48.64±0.85 | 33.17±0.64 | - |
| Meta-Learning | ProtoNet (Snell et al., 2017) | 21.32±0.37 | 29.58±0.57 | 55.32±0.88 | 52.94±0.81 | | | | | - |
| Meta-Learning | MatchingNet (Vinyals et al., 2016) | 20.65±0.29 | 27.37±0.51 | 54.88±0.90 | 46.86±0.88 | 35.89±0.51 | 30.77±0.47 | 49.86±0.79 | 32.70±0.60 | - |
| Meta-Learning | GNN (Garcia & Bruna, 2018) | 22.00±0.46 | 32.02±0.66 | 63.69±1.03 | 64.48±1.08 | 45.69±0.68 | 31.79±0.51 | 53.10±0.80 | 35.60±0.56 | - |
| Meta-Learning | FWT (Tseng et al., 2020) | 22.04±0.44 | 31.58±0.67 | 62.36±1.05 | 66.36±1.04 | 47.47±0.75 | 31.67±0.53 | 53.17±0.79 | 35.95±0.58 | - |
| Meta-Learning | LRP (Sun et al., 2021) | 22.11±0.10 | 30.94±0.30 | 54.99±0.50 | 59.23±0.50 | 48.29±0.51 | 32.78±0.39 | 54.83±0.56 | 37.49±0.43 | - |
| Meta-Learning | ATA (Wang & Deng, 2021) | 22.10±0.20 | 33.21±0.40 | 61.35±0.50 | 67.47±0.50 | 45.00±0.50 | 33.61±0.40 | 53.47±0.50 | 36.42±0.40 | - |
| Meta-Learning | AFA (Hu & Ma, 2022) | 22.92±0.20 | 33.21±0.30 | 63.12±0.50 | 76.49 | 46.86±0.50 | 34.05±0.60 | 54.04±0.60 | 36.76±0.40 | - |
| Transfer-Learning | STARTUP✗ (Phoo & Hariharan, 2021) | 23.09±0.43 | 32.66±0.60 | 75.93±0.80 | 63.88±0.84 | 48.87±0.81 | 38.01±0.73 | 31.79±0.61 | 41.24±0.75 | 4905.79K |
| Transfer-Learning | DynDistill✗ (Islam et al., 2021) | 23.38±0.43 | 34.66±0.58 | 82.14±0.78 | 73.14±0.84 | 49.28±1.11 | 40.60±1.15 | 34.77±0.98 | 42.51±1.11 | 4905.79K |
| Transfer-Learning | ConFeSS* (Das et al., 2022) | 23.67 | 33.46 | 65.51 | 76.49 | | | | | - |
| Transfer-Learning | **LoDC-9 (Ours)** | 22.47±0.41 | 32.08±0.58 | 71.04±0.91 | 80.16±0.81 | 38.48±0.78 | 30.74±0.75 | 55.99±0.88 | 38.87±0.74 | **27.91K (99.43%↓)** |
| Transfer-Learning | **LoDC-12 (Ours)** | 22.34±0.42 | 32.28±0.57 | 71.58±0.80 | 80.32±0.82 | 38.32±0.77 | 30.80±0.59 | 55.87±0.87 | 38.59±0.72 | **32.16K (99.34%↓)** |
| Transfer-Learning | **DC-9 (Ours)** | 22.34±0.41 | 33.03±0.60 | 76.55±0.82 | 83.60±0.79 | 39.49±0.78 | 32.13±0.63 | 57.42±0.88 | 41.72±0.80 | **187.85K (96.17%↓)** |
| Transfer-Learning | **DC-12 (Ours)** | 22.47±0.41 | 32.83±0.60 | 75.87±0.83 | 83.34±0.79 | 39.72±0.77 | 32.04±0.64 | 57.45±0.88 | 41.65±0.79 | **188.06K (96.17%↓)** |
| | | | | | 5-way-5-shot | | | | | |
| Meta-Learning | FT (Guo et al., 2020) | 27.01±0.44 | 27.01±0.44 | 80.84±0.56 | 84.00±0.56 | 58.10±0.78 | 44.39±0.66 | 72.92±0.66 | 53.26±0.56 | - |
| Meta-Learning | RelationNet (Sung et al., 2018) | - | - | - | - | 57.77±0.69 | 37.33 ± 0.68 | 63.32±0.76 | 44.00±0.60 | - |
| Meta-Learning | ProtoNet (Snell et al., 2017) | 24.72±0.43 | 24.72±0.43 | 42.49±0.58 | 76.92±0.67 | | | | | - |
| Meta-Learning | MatchingNet (Vinyals et al., 2016) | 22.62±0.36 | 22.62±0.36 | 33.96±0.54 | 68.00±0.68 | 63.16±0.77 | 38.99±0.64 | 63.16±0.77 | 46.53±0.68 | - |
| Meta-Learning | GNN (Garcia & Bruna, 2018) | 25.27±0.46 | 43.94±0.67 | 83.64±0.77 | 87.96±0.67 | 62.25±0.65 | 44.28±0.63 | 70.84±0.65 | 52.53±0.59 | - |
| Meta-Learning | FWT (Tseng et al., 2020) | 25.18±0.45 | 43.17±0.70 | 83.01±0.79 | 87.11±0.67 | 66.98±0.68 | 44.90±0.64 | 73.94±0.67 | 53.85±0.62 | - |
| Meta-Learning | LRP (Sun et al., 2021) | 24.53±0.30 | 44.14±0.40 | 77.14±0.40 | 86.15±0.40 | 64.44±0.48 | 46.20±0.46 | 74.45±0.47 | 54.46±0.46 | - |
| Meta-Learning | ATA (Wang & Deng, 2021) | 24.32±0.40 | 44.91±0.40 | 83.75±0.40 | 90.59±0.30 | 66.22±0.50 | 49.14±0.40 | 75.48±0.40 | 52.69±0.40 | - |
| Meta-Learning | AFA (Hu & Ma, 2022) | 25.02±0.20 | 46.01±0.40 | 85.58±0.40 | 88.06±0.30 | 68.25±0.50 | 49.28±0.50 | 76.21±0.50 | 54.26±0.40 | - |
| Transfer-Learning | STARTUP✗ (Phoo & Hariharan, 2021) | 26.94±0.44 | 47.22±0.61 | 82.29±0.45 | 93.02±0.45 | | 46.73±0.73 | 69.56±0.66 | 55.40±0.78 | 4905.79K |
| Transfer-Learning | DynDistill✗ (Islam et al., 2021) | 28.31±0.46 | 49.36±0.59 | 89.08±0.47 | 95.54±0.38 | 62.86±1.06 | 51.98±1.18 | 70.98±0.94 | 58.63±1.14 | 4905.79K |
| Transfer-Learning | ConFeSS* (Das et al., 2022) | 27.09±0.24 | 48.85±0.29 | 84.65±0.38 | 88.88±0.51 | | | | | - |
| Transfer-Learning | **LoDC-9 (Ours)** | 25.89±0.44 | 43.96±0.58 | 87.22±0.52 | 94.40±0.40 | 52.79±0.80 | 45.07±0.68 | 74.50±0.66 | 55.21±0.74 | **27.91K (99.43%↓)** |
| Transfer-Learning | **LoDC-12 (Ours)** | 26.28±0.45 | 43.76±0.57 | 87.44±0.53 | 94.18±0.41 | 52.52±0.81 | 44.45±0.68 | 74.42±0.66 | 55.05±0.74 | **32.16K (99.34%↓)** |
| Transfer-Learning | **DC-9 (Ours)** | 25.85±0.85 | 45.42±0.56 | 90.46±0.42 | 95.44±0.40 | 54.16±0.81 | 46.08±0.71 | 76.48±0.65 | 58.34±0.78 | **187.85K (96.17%↓)** |
| Transfer-Learning | **DC-12 (Ours)** | 26.83±0.43 | 45.24±0.58 | 90.82±0.41 | 95.37±0.39 | 53.93±0.83 | 46.15±0.72 | 76.28±0.65 | 58.40±0.77 | **188.06K (96.17%↓)** |

out as it optimizes all parameters except for the atom-coefficients, positioning it as a potentially more efficient fine-tuning strategy in terms of parameter updates.

Remarkably, by learning less than approximately 2% of the total parameters in the vanilla backbone, both DC-9 and DC-12 exceed the performance. This observation led us to question whether our method's superiority stemmed from its reduced parameter complexity, which may be more suited to the limited amount of unlabeled samples available for fine-tuning. To investigate this, we independently explore Low-Rank Adaptation (LoRA) (Hu et al., 2021), commonly employed for fine-tuning

Table 6: **5-way 1-shot and 5-shot** accuracies (%) on **ResNet-18 (ImageNet-1K)** fine-tuned with 20% of target unlabeled samples. Highlighted rows show performance when the atom-coefficients are fine-tuned while the spatial atoms are frozen, noting the higher parameter count when solely finetuning the coefficients.

| Method | Dataset | | | | | | | | # of Params. |
| --- | --- | --- | --- | --- | --- | --- | --- | --- | --- |
| | ChestX | ISIC | EuroSAT | CropDisease | CUB | Cars | Places | Plantae | |
| **5-way 1-shot** | | | | | | | | | |
| DC-9 | 22.16±0.42 | 36.03±0.65 | 81.35±0.73 | 89.98±0.69 | 47.89±0.85 | 36.99±0.72 | 66.61±0.91 | 49.22±0.86 | 192.34K |
| DC-9-Coeff | 21.64±0.41 | 35.33±0.68 | 85.98±0.70 | 92.50±0.74 | 42.15±0.86 | 36.04±0.76 | 69.68±0.97 | 48.26±0.90 | 11,176.51K |
| DC-12 | 22.12±0.42 | 36.48±0.65 | 81.42±0.74 | 89.72±0.69 | 48.05±0.86 | 36.86±0.75 | 66.90±0.90 | 49.02±0.87 | 192.77K |
| DC-12-Coeff | 21.60±0.42 | 35.70±0.69 | 85.78±0.73 | 92.44±0.72 | 41.63±0.85 | 36.26±0.78 | 69.79±0.95 | 47.79±0.90 | 14,838.34K |
| **5-way 5-shot** | | | | | | | | | |
| DC-9 | 26.50±0.45 | 49.02±0.62 | 93.81±0.30 | 97.60±0.30 | 70.05±0.80 | 54.86±0.77 | 83.44±0.55 | 67.74±0.79 | 192.34K |
| DC-9-Coeff | 24.52±0.42 | 47.68±0.60 | 95.32±0.27 | 97.62±0.37 | 56.26±0.87 | 48.78±0.84 | 84.76±0.53 | 65.52±0.84 | 11,176.51K |
| DC-12 | 26.40±0.46 | 50.14±0.64 | 93.90±0.31 | 97.61±0.30 | 70.65±0.78 | 55.34±0.78 | 83.60±0.55 | 67.41±0.81 | 192.77K |
| DC-12-Coeff | 24.61±0.43 | 47.90±0.60 | 95.12±0.28 | 97.55±0.35 | 55.40±0.88 | 48.93±0.82 | 84.54±0.52 | 65.12±0.84 | 14,838.34K |

large language models, and adapt it to fine-tune convolutional filters within our framework. With LoRA, updates to a pre-trained weight matrix $\mathbf{W} \in \mathbb{R}^{d_{\text{out}} \times d_{\text{in}}}$ are constrained to a low-rank representation $\mathbf{W} + \mathbf{\Delta W} = \mathbf{W} + \mathbf{B} \cdot \mathbf{C}$, where $\mathbf{B} \in \mathbb{R}^{d_{\text{out}} \times r}$, $\mathbf{C} \in \mathbb{R}^{r \times d_{\text{in}}}$, and $r \ll \min(d_{\text{out}}, d_{\text{in}})$. During training, $\mathbf{W}$ is fixed and does not receive gradient updates, while $\mathbf{C}$ and $\mathbf{B}$ are adjusted as trainable parameters. To adapt LoRA to convolutional filters, we interpret $\mathbf{K} \in \mathbb{R}^{n \times c \times l \times l}$ as $\mathbb{R}^{n \cdot l \times c \cdot l}$, enabling the learning of a low-rank update $\mathbf{\Delta K}$, thus inherently reducing the parameter complexity. We conduct the LoRA experiments at rank 3, which approximately matches the parameter complexity of our variants. The resulting superiority of our method compared to LoRA dispels our doubts about parameter complexity and underscores the effectiveness of selectively fine-tuning the spatial atoms while preserving the atom-coefficients, thus supporting our hypothesis. Moreover, this hints that, atom-coefficient preservation aids in maintaining the foundational knowledge encapsulated in them, an aspect where conventional fine-tuning and LoRA do not measure up. Additionally, in Appendix D, we draw parallels on how our proposition of finetuning the spatial atoms is similar to or different from LoRA-style adaptation methods.

Furthermore, in Table 3, we perform a similar analysis using a ResNet-50 backbone. However, unlike the analysis with ResNet-18 (Table 1), which only involved decomposition of the $3 \times 3$ filters, in ResNet-50 we apply LoRA to existing $1 \times 1$ convolution filters. We designate this variant as LoDC. This modification was prompted by our observation that decomposing only the $3 \times 3$ filters resulted in a mere reduction of only $\approx 48\%$ in the number of trainable parameters, due to the ResNet-50 architecture's predominance of $1 \times 1$ convolutional filters, *cf*. Table 2. For a particular LoDC variant we use the same rank for the LoRA as the number of decomposed atoms. By extending LoRA to the $1 \times 1$ filters, we were able to further decrease the parameter complexity by $\approx 98\%$. Moreover, this inculcation also demonstrates our method's compatibility with LoRA. For the stand-alone LoRA comparison we use the rank of 4 to match the parameter complexity of our methods (LoDC-9 and LoDC-12). The extended analysis with different LoRA ranks for the number reported in Table 1 and 3 are conducted in Appendix E.2. Further comparisons with other LoRA-style PEFT methods and ablations on the number of different filter atoms are provided in Appendix E.4. Moreover, our current analysis on the ResNet family of architecture is based on the previous study by Oh et al. (2022). Nonetheless, we also experiment with ConvNext (Liu et al., 2022) architecture in Appendix E.5

**Comparisons with Related Methods.** Moreover, we further analyze our method in the standard setting (Phoo & Hariharan, 2021; Oh et al., 2022), which uses 20% of the total target dataset $\boldsymbol{D}_{\text{T}}$ in $\boldsymbol{D}_{\text{U}}$, still less compared to what is generally used in SSL pre-training, on a ResNet-18 backbone pre-trained on ImageNet-1K. We draw comparisons with previously established works such as STARTUP (Phoo & Hariharan, 2021) and DynDistill (Islam et al., 2021). Although these methods assume access to the base dataset and optimize the entire backbone, which leads to their superior performance, we still provide a comparative perspective. In Table 4, we delineate the comparisons of methods LoDC, as done for the ResNet-50 backbone, and DC with 9 and 12 filter atoms against straightforward fine-tuning using SimCLR, as well as against LoRA at ranks 12, 9, and 3. We chose ranks 12 and 9 to correspond with the number of spatial filter atoms, and rank 3 was selected to match the parameter complexity of the DC variants (DC-9 and DC-12). From the comparisons, we observe performance that is comparable to or better than previous methods, specifically on the ChestX, ISIC, EuroSAT, and Crop datasets, despite their substantial task differences from the base

dataset (ImageNet-1K) (Oh et al., 2022). This is notable considering we fine-tune only a very small fraction of the parameters and do not assume access to the base dataset. Furthermore, in Table 5, We offer comparative analysis against other CD-FSL benchmarks within Meta-Learning (ML) and Transfer-Learning (TL) frameworks, utilizing a ResNet-10 backbone pre-trained on *mini*ImageNet for TL approaches. It is important to note that direct comparison between ML and TL methods is not feasible due to their distinct training strategies. However, we include these comparisons to provide a comprehensive overview of the performance standings across all methods. Once again, our methods either outperform prior (TL-based) methods or are at least comparable, while only fine-tuning a very minimal fraction of the parameters, *i.e.*, the spatial atoms. At the same time, we do not assume access to the base data during fine-tuning.

**Fine-tuning Atom-Coefficients?** In Section 3.3, we hypothesized that selectively fine-tuning spatial filter atoms should suffice for model adaptation to the target task, prompting the question of the impact of also fine-tuning atom-coefficients. Therefore, in Table 6 we consider finetuning only the atom coefficients and observe better accuracy and parameter efficiency trade-off for the case when tuning only the spatial atoms. Moreover, we also conduct ablations with finetuning the atom-coefficients together with the spatial-atoms. Considering atom-coefficients function as linear transformations, implemented as $1 \times 1$ convolutions (as detailed in (3) and Figure 3), we employ LoRA on the atom-coefficients, dubbing it as LoCo (LoRA-Coefficients), with the rank set to same as the number of atoms, on both ResNet-18 and ResNet-10 backbones. Our ablation studies, which involve fine-tuning the spatial atoms and fine-tuning the atom-coefficients using LoRA are provided in Appendix Table 9 and 10, indicating that additionally fine-tuning atom-coefficients does not significantly improve performance compared to just fine-tuning spatial atoms. This observation reinforces the idea that spatial atoms alone are able to efficiently learn the task-specific knowledge, supporting our hypothesis about the effectiveness of selective spatial atom fine-tuning.

**Ablation on Self-Supervised Methods.**

We use the constrastive SimCLR objective in the *Conform* phase due to its simplicity and theoretical backing (10). Nonetheless, we also experiment with a non-constrastive SSL method, BYOL (Grill et al., 2020). In Table 7, we compare BYOL with SimCLR on 5-way 1-shot and 5-way 5-shot evaluations, in the standard setup. From the table we observe better performance of BYOL on CUB and Cars and sub-optimal performance on Places and Plantae. Nonetheless, it is important to note that our method is agnostic to the choice of SSL objective, as it primarily serves to guide the model's alignment to the target domain. Although we primarily use SimCLR for evaluation, other SSL objectives can be employed within our framework.

Table 7: Comparison of **BYOL** (Grill et al., 2020) against **SimCLR** (Chen et al., 2020) performance on 5-way 1-shot and 5-way 5-shot evaluations in the standard setup.

| Method | | Dataset | | | |
|---|---|---|---|---|---|
| | | CUB | Cars | Places | Plantae |
| 5-way 1-shot | | | | | |
| SimCLR (Chen et al., 2020) | DC-9 | 47.89±0.85 | 36.99±0.72 | 66.61±0.91 | 49.22±0.86 |
| | DC-12 | 48.05±0.86 | 36.86±0.75 | 66.90±0.90 | 49.02±0.87 |
| BYOL (Grill et al., 2020) | DC-9 | 52.91±0.87 | 36.07±0.62 | 48.17±0.83 | 38.36±0.66 |
| | DC-12 | 53.70±0.90 | 36.29±0.62 | 47.75±0.84 | 37.78±0.70 |
| 5-way 5-shot | | | | | |
| SimCLR (Chen et al., 2020) | DC-9 | 70.58±0.78 | 55.10±0.77 | 82.76±0.55 | 64.36±0.79 |
| | DC-12 | 70.68±0.76 | 54.96±0.76 | 82.31±0.56 | 64.63±0.81 |
| BYOL (Grill et al., 2020) | DC-9 | 83.22±0.66 | 62.35±0.75 | 77.77±0.64 | 60.77±0.78 |
| | DC-12 | 83.45±0.67 | 62.82±0.76 | 77.75±0.59 | 60.19±0.80 |

## 5 CONCLUSION

In conclusion, our work introduces *Divide and Conform* a novel, efficient method for adapting pre-trained models to specialized target tasks under the constraints of limited unlabeled target data, without relying on extensive base datasets. By decomposing convolution filters into spatial filter atoms and their atom-coefficients, we recalibrate the pre-trained model to align with the target task through selective fine-tuning of spatial atoms. This approach allows adaptation to task-specific spatial variances while preserving channel combination knowledge in the atom-coefficients, addressing model transferability and generalization challenges, especially in scenarios with no labeled data and significant domain discrepancy. Our method's empirical evaluations in CD-FSL setting highlight its effectiveness and efficiency, offering notable improvements over existing approaches with a lean parameter update mechanism.

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

# A  CD-FSL AS A TEST-BET TO EVALUATE REPRESENTATION TRANSFERABILITY.

We use CDFSL as a test-bed to assess model transferability that addresses substantial domain discrepancies between source and target in both input (diverse images domains) and label spaces (non-overlapping labels), presenting a more complex challenge than typical domain adaptation where the label space is shared or FSL with meta-datasets containing images from similar domains. CD-FSL ssumes scarcity of high-quality samples, underscoring the evaluation in the few-shot setting. In the typical self-supervised pre-training, linear probing—applying a linear classifier to a frozen backbone—is commonly used to evaluate representation quality. CD-FSL makes this setting stringent by linear-probing on a limited number of examples across 600 episodes over 8 existing, highly diverse, standard benchmark datasets offering a more rigorous assessment of the representation transferred from the base dataset. Therefore, we select CD-FSL as our test-bed for its comprehensive and stringent evaluation protocol.

# B  SIMCLR MOTVATION

The principal motivation for adopting SimCLR for fine-tuning is twofold.

Firstly, SimCLR stands out as one of the most straightforward yet effective SSL methods. The fundamental concept involves positioning semantically similar examples closer together in the representational space, while ensuring that dissimilar examples are kept apart. Formally, each instance in the current batch $\{\mathbf{I}_i\}_{i=1}^B$ of size $B$ undergoes augmentation to create an enhanced batch $\{\tilde{\mathbf{I}}_{2i-1}, \tilde{\mathbf{I}}_{2i}\}_{i=1}^B$, in which $\tilde{\mathbf{I}}_{2i-1}$ and $\tilde{\mathbf{I}}_{2i}$ represent distinct augmentations derived from the same input $\mathbf{I}_i$. Subsequently, the transformed representations $\{\mathbf{z}_{2i-1} = \boldsymbol{h} \circ \boldsymbol{f}(\tilde{\mathbf{I}}_{2i-1}), \mathbf{z}_{2i} = \boldsymbol{h} \circ \boldsymbol{f}(\tilde{\mathbf{I}}_{2i})\}_{i=1}^B$ are extracted from the backbone network $\boldsymbol{f}$ equipped with projection layers $\boldsymbol{h}$. Utilizing these representations, SimCLR conducts contrastive learning to minimize the contrastive loss given by:

$$\mathcal{L}_{\text{SimCLR}} = \frac{1}{2B} \sum_{i=1}^B \left[ \ell(2i-1, 2i) + \ell(2i, 2i-1) \right], \tag{8}$$

where

$$\ell(i, j) = -\log \left( \frac{\exp(\text{sim}(\mathbf{z}_i, \mathbf{z}_j)/\tau)}{\sum_{n=1}^{2B} \mathbf{1}_{[n \neq i]} \exp(\text{sim}(\mathbf{z}_i, \mathbf{z}_n)/\tau)} \right). \tag{9}$$

Here, $\mathbf{1}$ is an indicator function, $\tau$ is a temperature hyperparameter, and $\text{sim}(\mathbf{u}, \mathbf{v}) = \frac{\mathbf{u}^\top \mathbf{v}}{\|\mathbf{u}\| \cdot \|\mathbf{v}\|}$ measures cosine similarity between two vectors $\mathbf{u}$ and $\mathbf{v}$. The objective (9) is also know as NT-Xent loss (normalized temperature-scaled cross-entropy loss) (Chen et al., 2020).

Secondly, Cao et al. (2021) established the following bound:

$$\mathcal{L}_{\text{sup}} \leq \gamma_0 \mathcal{L}_U^- + \gamma_1 s(\boldsymbol{f}_k). \tag{10}$$

Within this bound, $\mathcal{L}_{\text{sup}}$ signifies the metric for assessing supervised learning representations. The metric $\mathcal{L}_U^-$ evaluates unsupervised contrastive samples, specifically those that are true negatives. The function $s(\boldsymbol{f}_k)$ quantifies the variation within classes as interpreted by the principal encoder $\boldsymbol{f}_k$ (He et al., 2020). The constants $\gamma_0$ and $\gamma_1$ vary in accordance with class distribution. The term $\mathcal{L}_{\text{sup}}$ can represent the training loss for supervised few-shot meta-learning techniques that are adaptable to new classes. Given that $\mathcal{L}_U^-$ sets the upper boundary for $\mathcal{L}_{\text{sup}}$, any minimization on $\mathcal{L}_U^-$ naturally lowers $\mathcal{L}_{\text{sup}}$. It is also noteworthy that $\mathcal{L}_U^-$ can be minimized without constraints since it is assessed solely through true negative samples, rendering contrastive methods advantageous for learning representations conducive to efficient few-shot learning.

# C  DATASET DESCRIPTION

Our analysis utilizes two base datasets: *mini*ImageNet (Vinyals et al., 2016) (a subset of ImageNet-1K (Deng et al., 2009)) and the complete ImageNet-1K, along with eight target datasets from CD-FSL benchmarks from BSCD-FSL (Guo et al., 2020) and FWT (Tseng et al., 2020). BSCD-FSL

Table 8: Dataset Summary: Indicating the number of classes, total number of samples, and the number of samples for the 1%, 5%, and 20% (standard) splits used for unsupervised fine-tuning.

| Datasets | ChestX | ISIC | EuroSAT | Crop | CUB | Cars | Places | Plantae |
|---|---|---|---|---|---|---|---|---|
| # of classes | 7 | 7 | 10 | 38 | 200 | 196 | 16 | 69 |
| # of samples | 25,848 | 10,015 | 27,000 | 43,456 | 11,788 | 16,185 | 27,440 | 26,650 |
| 1% of samples | 258 | 100 | 270 | 434 | 117 | 161 | 274 | 266 |
| 5% of samples | 1,292 | 500 | 1,350 | 2,172 | 589 | 809 | 1,372 | 1,332 |
| 20% of samples (standard setup Oh et al. (2022)) | 5,169 | 2,003 | 5,400 | 8,691 | 2,357 | 3,237 | 5,488 | 5,330 |

encompasses specialized datasets such as CropDisease (Crop) (Mohanty et al., 2016), which features imagery of plant diseases; EuroSAT (Helber et al., 2019), providing satellite imagery of various landscapes; ISIC (Codella et al., 2018), with dermatoscopic images of skin lesions; and ChestX Wang et al. (2017b), containing X-ray images of the human thoracic region. It is crucial to acknowledge the significant task and domain disparities between these datasets and the base datasets. Additionally, we incorporate four more datasets prominently utilized in CD-FSL literature, from FWT, for more fine-grained analysis: Places (Zhou et al., 2018), Plantae (Van Horn et al., 2018), Cars (Krause et al., 2013), and CUB (Wah et al., 2011). All datasets follow the standard splitting proposed by Oh et al. (2022). For all our experiments we resize the images to $224 \times 224$ to be consistent across all the datasets for base pre-training and target fine-tuning.

The datasets are summarized as follows:

- **CropDisease (Crop) (Mohanty et al., 2016):** Features images of plant diseases.

- **EuroSAT (Helber et al., 2019):** A collection of satellite images depicting various landscapes.

- **ISIC (Codella et al., 2018):** Consists of dermatoscopic images of skin lesions.

- **ChestX (Wang et al., 2017b):** Comprises X-ray images of the chest area.

For the datasets Places, Plantae, Cars, and CUB, due the lack of a standardized approach for dividing training and evaluation sets, we adopt the sampling strategy adopted by Oh *et al*. Oh et al. (2022), aiming to make the datasets more manageable for FSL applications while ensuring they remain representative:

- **Places (Zhou et al., 2018):** Contains scene recognition images, with a subset of 16 out of 365 classes sampled.

- **Plantae (Van Horn et al., 2018):** Features images of plants, with the top 69 classes selected from 2,917 due to class imbalance.

- **Cars (Krause et al., 2013):** Includes images of 196 car models, from the train, test and val split.

- **CUB (Wah et al., 2011):** Contains images of 200 bird species, from the train, test and val split.

Table 8 provides an overall statistics of the target datasets.

## D   DC vs. LoRA

Our method focuses solely on fine-tuning spatial atoms. Unlike LoRA-style adaptation (Hu et al., 2021), which learns a residual component $\Delta \mathbf{K}$ across the entire filter ($\mathbf{K}_t = \mathbf{K}_b + \Delta \mathbf{K}$), our method fine-tunes from $\mathbf{K}_b = \mathbf{A} \mathbf{D}_b$ to $\mathbf{K}_t = \mathbf{A} \mathbf{D}_t$, expressed as $\mathbf{K}_t = \mathbf{A}(\mathbf{D}_b + \Delta \mathbf{D})$ where $\Delta \mathbf{D} = \mathbf{D}_t - \mathbf{D}_b$. However, we avoid LoRA's residual-style fine-tuning of atoms ($\mathbf{D}$), which requires maintaining both $\mathbf{D}_b$ and $\Delta \mathbf{D}$ in memory, opting instead for a direct transition from $\mathbf{D}_b$ to $\mathbf{D}_t$ to reduce memory usage. This is how our method relates/differs from LoRA.

# E  ADDITIONAL EXPERIMENTS

## E.1  EXTENDED RESULTS ON ATOM-COEFFICIENT FINE-TUNING

Table 9: **5-way 1-shot** CD-FSL performance (%) comparisons with 95% confidence intervals using **ResNet-10 (*mini*ImageNet)** and **ResNet-18 (ImageNet-1K)** backbones across different DC variants. The last column indicates the number of parameters fine-tuned for each variant. Here we compare our methods by also fine-tuning the atom-coefficients using LoRA (LoCO).

| Method | Dataset | | | | | | | | # of Params. |
|---|---|---|---|---|---|---|---|---|---|
| | ChestX | ISIC | EuroSAT | Crop | CUB | Cars | Places | Plantae | |
| ResNet-10 | | | | | | | | | |
| LoDC-9 | 22.47±0.41 | 32.08±0.58 | 71.04±0.91 | 80.16± 0.81 | 38.48±0.78 | 30.74±0.75 | 55.99±0.88 | 38.87±0.74 | 27.91K |
| LoDC-12 | 22.34±0.42 | 32.28±0.57 | 71.58±0.88 | 79.70±0.82 | 38.32±0.77 | 30.80±0.59 | 55.87±0.87 | 38.59±0.72 | 32.16K |
| DC-9 | 22.34±0.41 | 33.03±0.60 | 76.55±0.82 | 83.60±0.79 | 39.49±0.78 | 32.13±0.63 | 57.42±0.88 | 41.72±0.80 | 187.85K |
| DC-12 | 22.47±0.41 | 32.83±0.60 | 75.87.0.83 | 83.34±0.79 | 39.72±0.77 | 32.04±0.64 | 57.45±0.88 | 41.65±0.79 | 188.06K |
| LoDC-9 w/ LoCo | 22.32±0.41 | 32.34±0.58 | 71.34±0.89 | 79.89±0.81 | 38.61±0.77 | 30.80±0.60 | 55.89±0.86 | 38.94±0.73 | 164.42K |
| LoDC-12 w/ LoCo | 22.46±0.41 | 32.33±0.58 | 71.60±0.88 | 79.80±0.80 | 38.46±0.46 | 30.77±0.59 | 56.13±0.87 | 38.80±0.73 | 267.17K |
| DC-9 w/ LoCo | 22.55±0.41 | 33.49±0.61 | 76.12±0.82 | 83.48±0.79 | 39.37±0.76 | 32.23±0.63 | 57.16±0.86 | 42.00±0.79 | 324.36K |
| DC-12 w/ LoCo | 22.38±0.40 | 33.03±0.60 | 76.33±0.83 | 83.45±0.79 | 38.28±0.75 | 31.97±0.64 | 57.25±0.88 | 41.94±0.79 | 423.07K |
| ResNet-18 | | | | | | | | | |
| LoDC-9 | 22.36±0.42 | 33.61±0.61 | 77.64±0.78 | 85.37±0.74 | 48.07±0.86 | 35.98±0.68 | 65.47±0.89 | 46.41±0.81 | 32.40K |
| LoDC-12 | 22.47±0.41 | 33.95±0.60 | 77.25±0.79 | 85.41±0.74 | 48.38±0.86 | 36.28±0.71 | 66.34±0.88 | 46.75±0.83 | 36.86K |
| DC-9 | 22.16±0.42 | 36.03±0.65 | 81.35±0.73 | 89.98±0.69 | 47.89±0.85 | 36.99±0.72 | 66.61±0.91 | 49.22±0.86 | 192.34K |
| DC-12 | 22.12±0.42 | 36.48±0.65 | 81.42±0.74 | 89.72±0.69 | 48.05±0.86 | 36.86±0.75 | 66.90±0.90 | 49.02±0.87 | 192.77K |
| LoDC-9 w/ LoCo | 22.26±0.42 | 34.00±0.60 | 76.64±0.80 | 85.54±0.73 | 48.03±0.86 | 36.08±0.71 | 65.53±0.89 | 46.56±0.83 | 341.71K |
| LoDC-12 w/ LoCo | 22.44±0.42 | 33.91±0.60 | 78.06±0.76 | 84.83±0.74 | 48.11±0.85 | 35.89±0.71 | 65.68±0.89 | 46.25±0.81 | 571.39K |
| DC-9 w/ LoCo | 22.27±0.42 | 36.04±0.65 | 81.40±0.74 | 90.00±0.68 | 48.04±0.83 | 37.32±0.73 | 66.86±0.89 | 48.89±0.85 | 501.65K |
| DC-12 w/ LoCo | 21.99±0.40 | 36.13±0.66 | 80.83±0.74 | 89.71±0.69 | 48.08±0.84 | 37.24±0.73 | 67.18±0.92 | 48.86±0.86 | 727.30K |

In Table 9, we additionally provide the results for 5-way 1-shot experiments on all our DC variants, both with and without atom-coefficient fine-tuning. Furthermore, we offer a comprehensive summary of all our variants.

- **LoDC-$X$:** This variant employs LoRA (Hu et al., 2021) with a rank of $X$ on the $1 \times 1$ filters and our proposed decomposition on the $3 \times 3$ convolution filters into $X$ spatial atoms and their corresponding atom-coefficients. The remainder of the convolution filters and parameters are left unchanged. In this variant, only the atom-coefficients are kept static.

- **DC-$X$:** This variant decomposes the $3 \times 3$ convolution filters into $X$ spatial atoms and their corresponding atom-coefficients. Similar to the LoDC-$X$ variant, and only the atom-coefficients are kept static.

As discussed in the main manuscript regarding fine-tuning the atom-coefficients, we created additional variants that also fine-tune the atom-coefficients. It is important to note that operating the atom-coefficients as $1 \times 1$ convolutions and fine-tuning the entire atom-coefficients along with the spatial atoms would be equivalent to fully fine-tuning the model. Therefore, to maintain parameter efficiency, we apply LoRA to the atom coefficients.

- **LoDC-$X$ w/ LoCo:** This variant is similar to the LoDC-$X$ variant mentioned above, but with the addition of fine-tuning the atom-coefficients using LoRA with a rank of $X$.

- **DC-$X$ w/ LoCo:** This variant is similar to the DC-$X$ variant mentioned above, but with the addition of fine-tuning the atom-coefficients using LoRA with a rank of $X$.

## E.2  EXTENDED ANALYSIS WITH FEW UNLABELED SAMPLES

Considering the parameter efficiency of our proposed methods (DC-9 and DC-12), we investigate whether their superiority stems from reduced parameter complexity, potentially better suited for the limited amount of unlabeled samples available for fine-tuning. To further analyze this, we extend the results from Tables 1 and 3 in the main manuscript by conducting additional experiments using LoRA at lower parameter complexities. We incorporate LoRA performances at rank 2 for ResNet-18 (*cf*. Table 11) and at ranks 3 and 5 for ResNet-50 (*cf*. Table 12). Our observations indicate that merely reducing the parameter complexity does not enhance the few-shot performance.

Table 10: **5-way 5-shot** CD-FSL performance (%) comparisons with 95% confidence intervals using **ResNet-10 (*mini*ImageNet)** and **ResNet-18 (ImageNet-1K)** backbones across different DC variants. The last column indicates the number of parameters fine-tuned for each variant. Here we compare our methods by also fine-tuning the atom-coefficients using LoRA (LoCO).

| Method | ChestX | ISIC | EuroSAT | Crop | CUB | Cars | Places | Plantae | # of Params. |
|---|---|---|---|---|---|---|---|---|---|
| | | | | | Dataset | | | | |
| | | | | | ResNet-10 | | | | |
| LoDC-9 | 25.89±0.44 | 43.96±0.58 | 87.22±0.52 | 94.40±0.40 | 52.79±0.80 | 45.07±0.68 | 74.50±0.66 | 55.21±0.74 | 27.91K |
| LoDC-12 | 26.28±0.45 | 43.76±0.57 | 87.44±0.53 | 94.18±0.41 | 52.52±0.81 | 44.45±0.68 | 74.42±0.66 | 55.05±0.74 | 32.16K |
| DC-9 | 25.85±0.85 | 45.42±0.56 | 90.46±0.42 | 95.44±0.40 | 54.16±0.81 | 46.08±0.71 | 76.48±0.65 | 58.34±0.78 | 187.85K |
| DC-12 | 26.83±0.43 | 45.24±0.58 | 90.82±0.41 | 95.37±0.39 | 53.93±0.83 | 46.15±0.72 | 76.28±0.65 | 58.40±0.77 | 188.06K |
| LoDC-9 w/ LoCo | 25.80±0.43 | 44.34±0.58 | 87.26±0.52 | 94.23±0.42 | 52.93±0.80 | 45.08±0.68 | 74.06±0.67 | 55.09±0.73 | 164.42K |
| LoDC-12 w/ LoCo | 26.49±0.44 | 44.09±0.58 | 87.50±0.52 | 94.19±0.42 | 52.72±0.80 | 44.22±0.68 | 74.55±0.67 | 55.15±0.74 | 267.17K |
| DC-9 w/ LoCo | 26.04±0.45 | 46.13±0.59 | 90.76±0.42 | 95.38±0.40 | 54.06±0.81 | 46.16±0.71 | 76.35±0.64 | 58.52±0.79 | 324.36K |
| DC-12 w/ LoCo | 26.71±0.45 | 45.37±0.58 | 90.81±0.42 | 95.48±0.39 | 54.08±0.81 | 46.37±0.70 | 76.47±0.64 | 58.48±0.77 | 423.07K |
| | | | | | ResNet-18 | | | | |
| LoDC-9 | 26.65±0.44 | 46.20±0.60 | 91.55±0.39 | 96.54±0.33 | 70.58±0.78 | 55.10±0.77 | 82.76±0.55 | 64.36±0.79 | 32.40K |
| LoDC-12 | 26.60±0.45 | 46.82±0.60 | 91.54±0.38 | 96.49±0.34 | 70.68±0.76 | 54.96±0.76 | 82.31±0.56 | 64.63±0.81 | 36.86K |
| DC-9 | 26.50±0.45 | 49.02±0.62 | 93.81±0.30 | 97.60±0.30 | 70.05±0.80 | 54.86±0.77 | 83.44±0.55 | 67.74±0.79 | 192.34K |
| DC-12 | 26.40±0.46 | 50.14±0.64 | 93.90±0.31 | 97.61±0.30 | 70.65±0.78 | 55.34±0.78 | 83.60±0.55 | 67.41±0.81 | 192.77K |
| LoDC-9 w/ LoCo | 26.91±0.46 | 46.47±0.61 | 91.53±0.35 | 96.48±0.33 | 69.98±0.78 | 55.08±0.76 | 82.78±0.55 | 64.51±0.80 | 341.71K |
| LoDC-12 w/ LoCo | 26.77±0.45 | 46.70±0.62 | 91.91±0.37 | 96.38±0.34 | 70.24±0.78 | 55.32±0.77 | 82.80±0.58 | 64.34±0.79 | 571.39K |
| DC-9 w/ LoCo | 26.32±0.44 | 49.95±0.63 | 94.14±0.31 | 97.64±0.29 | 70.00±0.82 | 55.21±0.80 | 83.49±0.55 | 67.51±0.80 | 501.65K |
| DC-12 w/ LoCo | 26.53±0.46 | 49.70±0.62 | 93.93±0.32 | 97.63±0.30 | 70.67±0.80 | 54.87±0.79 | 83.64±0.55 | 67.51±0.80 | 727.30K |

Table 11: Extended results on **5-way 1-shot and 5-shot** accuracy (%) with 95% confidence using **ResNet-18 (ImageNet-1K)**, detailing unlabeled data use (first column) and parameters fine-tuned (last column). Best and second-best results are highlighted in **maroon** and **navy**.

| % Unlabeled Data | Method | ChestX | ISIC | EuroSAT | Crop | CUB | Cars | Places | Plantae | # of Params. |
|---|---|---|---|---|---|---|---|---|---|---|
| | | | | | | Dataset | | | | |
| | | | | | | 5-way 1-shot | | | | |
| 1% | SimCLR (Chen et al., 2020) | 22.20±0.41 | 32.19±0.58 | 70.76±0.90 | 82.56±0.86 | 44.22±0.85 | 32.71±0.67 | 59.85±0.91 | 41.48±0.82 | 11,176.51K |
| | LoRA-3 (Hu et al., 2021) | 22.38±0.42 | 31.65±0.54 | 69.96±0.89 | 83.36±0.84 | 44.41±0.85 | 32.60±0.65 | 59.07±0.92 | 40.71±0.78 | 218.75K |
| | LoRA-2 (Hu et al., 2021) | 22.15±0.41 | 31.45±0.53 | 69.17±0.89 | 83.38±0.85 | 43.43±0.85 | 32.42±0.66 | 59.24±0.94 | 40.87±0.81 | 149.03K |
| | **DC-9 (Ours)** | 22.87±0.48 | 33.73±0.59 | 69.10±0.85 | 82.35±0.79 | 50.61±0.88 | 34.42±0.67 | 59.50±0.86 | 42.92±0.77 | 192.34K |
| | **DC-12 (Ours)** | 22.62±0.42 | 34.34±0.59 | 68.88±0.83 | 83.28±0.81 | 49.19±0.86 | 33.61±0.66 | 59.98±0.88 | 43.31±0.81 | 192.77K |
| 5% | SimCLR (Chen et al., 2020) | 21.86±0.41 | 34.59±0.60 | 81.92±0.77 | 89.66±0.76 | 42.13±0.86 | 35.11±0.73 | 64.76±0.91 | 45.20±0.85 | 11,176.51K |
| | LoRA-3 (Hu et al., 2021) | 21.86±0.41 | 35.20±0.64 | 81.90±0.76 | 91.68±0.69 | 41.84±0.84 | 34.38±0.70 | 65.26±0.92 | 45.16±0.84 | 218.75K |
| | LoRA-2 (Hu et al., 2021) | 21.85±0.41 | 34.87±0.62 | 81.83±0.79 | 90.45±0.72 | 42.25±0.82 | 34.71±0.71 | 65.08±0.93 | 45.43±0.85 | 149.03K |
| | **DC-9 (Ours)** | 22.45±0.42 | 37.20±0.65 | 77.24±0.80 | 88.47±0.70 | 46.34±0.84 | 35.91±0.70 | 64.07±0.86 | 46.29±0.82 | 192.34K |
| | **DC-12 (Ours)** | 22.48±0.42 | 35.73±0.61 | 77.88±0.76 | 88.60±0.70 | 49.48±0.86 | 36.06±0.70 | 64.27±0.84 | 46.02±0.82 | 192.77K |
| | | | | | | 5-way 5-shot | | | | |
| 1% | SimCLR Chen et al. (2020) | 25.01±0.42 | 42.14±0.55 | 84.62±0.54 | 92.96±0.48 | 63.27±0.81 | 47.50±0.76 | 76.34±0.62 | 58.19±0.79 | 11,176.51K |
| | LoRA-3 (Hu et al., 2021) | 24.79±0.43 | 41.93±0.57 | 84.76±0.54 | 93.33±0.46 | 63.13±0.81 | 47.46±0.75 | 75.92±0.64 | 57.27±0.77 | 218.75K |
| | LoRA-2 (Hu et al., 2021) | 24.84±0.43 | 41.40±0.54 | 84.94±0.55 | 93.24±0.47 | 61.97±0.81 | 47.17±0.78 | 76.46±0.63 | 57.37±0.76 | 149.03K |
| | **DC-9 (Ours)** | 26.63±0.43 | 45.65±0.60 | 86.65±0.48 | 94.42±0.39 | 73.74±0.76 | 52.28±0.75 | 80.09±0.58 | 61.23±0.76 | 192.34K |
| | **DC-12 (Ours)** | 26.47±0.46 | 46.35±0.57 | 86.34±0.49 | 94.64±0.41 | 71.89±0.78 | 51.25±0.75 | 80.46±0.58 | 61.92±0.75 | 192.77K |
| 5% | SimCLR Chen et al. (2020) | 25.01±0.42 | 46.71±0.57 | 92.51±0.38 | 96.72±0.36 | 58.48±0.81 | 48.60±0.81 | 80.17±0.58 | 61.90±0.76 | 11,176.51K |
| | LoRA-3 (Hu et al., 2021) | 24.92±0.42 | 46.15±0.56 | 92.36±0.37 | 96.74±0.38 | 58.99±0.80 | 48.22±0.80 | 80.40±0.57 | 61.64±0.78 | 218.75K |
| | LoRA-2 (Hu et al., 2021) | 24.89±0.43 | 46.59±0.59 | 92.40±0.39 | 97.01±0.34 | 58.68±0.81 | 47.91±0.77 | 79.86±0.59 | 61.91±0.77 | 149.03K |
| | **DC-9 (Ours)** | 26.70±0.45 | 49.96±0.57 | 91.75±0.38 | 97.32±0.29 | 70.25±0.76 | 54.21±0.78 | 81.87±0.55 | 64.63±0.80 | 192.34K |
| | **DC-12 (Ours)** | 26.67±0.44 | 48.95±0.57 | 91.92±0.38 | 97.37±0.28 | 70.12±0.76 | 53.71±0.79 | 82.10±0.56 | 64.62±0.79 | 192.77K |

## E.3 SENSITIVITY TO THE CHOICE OF SPARSITY COEFFICIENT ($\lambda$)

We conduct ablation studies to assess the impact of different value of sparsity coefficient, $\lambda$, applied to $\mathbf{A}$ during the *Divide* step for decomposing $\mathbf{K}$. In Table 14, we present the 5-way 1-shot and 5-shot performances for our proposed variants, DC-9 and DC-12, with $\lambda$ values set to $\{10^{-4}, 10^{-6}, 10^{-8}\}$. The results indicate that the few-shot performance target is robust across different $\lambda$ values for a given variant, exhibiting minimal variance. This robustness might stem from the compensation provided by the spatial atoms to the atom coefficients during the time of unsupervised fine-tuning.

## E.4 ABLATION STUDY ON DIFFERENT NUMBER OF FILTER ATOMS, AND PEFT METHODS

In Table 15, we conduct the ablation study on the number of decomposed filter atoms and observe minute variabilities in the downstream few-shot accuracies.

In Table 13, we provided the decomposition residual error that we observed for different number of filter atoms, we observed a notable reduction in the decomposition error for filter atoms 9 and beyond andthere fore we limited our analysis to 9 and 12.

Table 12: Extended results on **5-way 1-shot and 5-shot** accuracy (%) with 95% confidence using **ResNet-50 (ImageNet-1K)**, detailing unlabeled data use (first column) and parameters fine-tuned (last column). Best and second-best results are highlighted in **maroon** and **navy**.

| % Unlabeled Data | Method | Dataset | | | | | | | | # of Params. |
|---|---|---|---|---|---|---|---|---|---|---|
| | | ChestX | ISIC | EuroSAT | Crop | CUB | Cars | Places | Plantae | |
| | | | | | **5-way 1-shot** | | | | | |
| 1% | SimCLR (Chen et al., 2020) | 22.14±0.40 | 30.47±0.52 | **72.56±0.88** | 84.32±0.86 | 39.59±0.80 | 30.49±0.64 | 58.22±0.93 | 43.21±0.84 | 23,508.03K |
| | LoRA-3 (Hu et al., 2021) | 21.81±0.42 | 31.42±0.59 | 72.28±0.88 | 83.95±0.87 | 46.01±0.85 | 32.60±0.65 | 59.87±0.92 | 41.40±0.83 | 391.29K |
| | LoRA-4 (Hu et al., 2021) | 22.41±0.42 | 31.17±0.57 | 71.49±0.86 | **84.66±0.83** | 45.06±0.86 | 28.77±0.61 | 60.03±0.94 | 42.10±0.81 | 504.01K |
| | LoRA-5 (Hu et al., 2021) | 22.41±0.43 | 31.21±0.58 | 72.55±0.87 | 84.38±0.84 | 40.18±0.82 | 32.09±0.63 | 59.69±0.96 | 41.22±0.81 | 616.74K |
| | **LoDC-9 (Ours)** | **22.56±0.44** | **34.42±0.60** | 71.61±0.83 | 84.04±0.80 | 52.44±0.85 | 32.90±0.66 | 63.54±0.88 | 44.39±0.81 | **437.07K** |
| | **LoDC-12 (Ours)** | 22.61±0.43 | 33.40±0.59 | 71.64±0.81 | 84.09±0.81 | 53.33±0.87 | 34.97±0.68 | 63.67±0.88 | 44.28±0.80 | 561.92K |
| 5% | SimCLR (Chen et al., 2020) | 21.81±0.41 | **35.62±0.65** | **85.53±0.74** | 91.59±0.69 | 41.52±0.85 | 34.55±0.72 | **68.01±0.91** | 46.16±0.88 | 23,508.03K |
| | LoRA-3 (Hu et al., 2021) | **22.00±0.41** | 35.18±0.65 | 83.10±0.77 | **91.68±0.69** | 36.04±0.77 | 32.62±0.71 | 67.86±0.89 | 46.10±0.87 | 391.29K |
| | LoRA-4 (Hu et al., 2021) | 21.91±0.40 | 34.42±0.60 | 83.61±0.73 | 91.33±0.71 | 40.63±0.83 | **35.61±0.73** | 67.70±0.92 | 47.21±0.88 | 504.01K |
| | LoRA-5 (Hu et al., 2021) | 21.98±0.41 | 34.57±0.61 | 83.87±0.74 | **91.97±0.70** | 39.93±0.82 | 34.64±0.73 | 67.36±0.91 | 47.15±0.89 | 616.74K |
| | **LoDC-9 (Ours)** | 21.95±0.42 | 35.05±0.61 | 79.57±0.73 | 88.84±0.71 | 50.48±0.89 | 35.20±0.70 | 68.06±0.84 | 48.13±0.85 | **437.07K** |
| | **LoDC-12 (Ours)** | 22.22±0.43 | 35.37±0.61 | 80.53±0.76 | 88.09±0.71 | 51.14±0.90 | 34.84±0.73 | 67.78±0.82 | 48.07±0.86 | 561.92K |
| | | | | | **5-way 5-shot** | | | | | |
| 1% | SimCLR (Chen et al., 2020) | 24.55±0.41 | 40.17±0.54 | 86.04±0.51 | 93.64±0.46 | 54.95±0.81 | 42.74±0.71 | 75.56±0.63 | 60.13±0.79 | 23,508.03K |
| | LoRA-3 (Hu et al., 2021) | 23.86±0.42 | 41.18±0.57 | 86.02±0.50 | 93.13±0.59 | 64.45±0.77 | 46.99±0.71 | 76.82±0.62 | 57.69±0.78 | 391.29K |
| | LoRA-4 (Hu et al., 2021) | 24.86±0.43 | 40.97±0.59 | 85.96±0.48 | 93.78±0.44 | 63.36±0.79 | 38.62±0.68 | 76.56±0.65 | 59.18±0.77 | 504.01K |
| | LoRA-5 (Hu et al., 2021) | 25.03±0.43 | 39.73±0.54 | 86.47±0.48 | 93.60±0.45 | 56.38±0.81 | 45.30±0.72 | 76.71±0.62 | 57.35±0.78 | 616.74K |
| | **LoDC-9 (Ours)** | **25.60±0.43** | **46.40±0.57** | 87.46±0.48 | 94.62±0.42 | 74.72±0.77 | 48.16±0.76 | 82.02±0.57 | 63.00±0.77 | **437.07K** |
| | **LoDC-12 (Ours)** | 25.47±0.44 | 45.63±0.57 | 87.63±0.47 | 94.47±0.42 | 75.98±0.76 | 52.36±0.77 | 82.20±0.58 | 62.95±0.78 | 561.92K |
| 5% | SimCLR (Chen et al., 2020) | 25.04±0.42 | 46.50±0.59 | 93.04±0.36 | **97.59±0.31** | 57.65±0.81 | 47.04±0.77 | 82.38±0.55 | 63.12±0.80 | 23,508.03K |
| | LoRA-3 (Hu et al., 2021) | 24.82±0.41 | 47.24±0.61 | 92.79±0.37 | **97.60±0.31** | 48.20±0.79 | 43.08±0.76 | 82.29±0.56 | 62.46±0.79 | 391.29K |
| | LoRA-4 (Hu et al., 2021) | 24.94±0.42 | 46.20±0.59 | **93.18±0.35** | 97.34±0.32 | 55.76±0.82 | 48.83±0.79 | 81.83±0.55 | 64.23±0.79 | 504.01K |
| | LoRA-5 (Hu et al., 2021) | 25.08±0.43 | 45.15±0.61 | **93.35±0.34** | 97.51±0.31 | 54.50±0.83 | 47.24±0.81 | 81.86±0.57 | 63.47±0.78 | 616.74K |
| | **LoDC-9 (Ours)** | **25.68±0.44** | 48.00±0.58 | 92.34±0.37 | 97.31±0.29 | 71.20±0.78 | 50.69±0.76 | 85.05±0.51 | 67.00±0.77 | **437.07K** |
| | **LoDC-12 (Ours)** | 25.89±0.42 | 48.57±0.60 | 92.72±0.37 | 97.08±0.30 | 71.98±0.76 | 50.43±0.78 | 85.03±0.50 | 66.78±0.79 | 561.92K |

Table 13: **AD** decomposition error of the **ResNet-18 (ImageNet-1K)** base model's pretrained filter.

| Method | DC-6 | DC-9 | DC-12 | DC-15 |
|---|---|---|---|---|
| **AD** Error | 2.0341e-07 | 5.2200e-09 | 4.6879e-09 | 4.9798e-09 |

In Table 16, we compare additional PEFT methods, LoHA (Low-rank Hadamard Adaptation) and LoKrA (Low-rank Kronecker Adaptation) (Yeh et al., 2024) under similar parameter-complexity.

### E.5 ANALYSIS WITH CONVNEXT

In Table 17, we applied our decomposition strategy to the ConvNext architecture (Liu et al., 2022). However, given the consistently lower performance across baselines compared to ResNet-18, we limited our further evaluations to the ResNet family. Despite this, our method still demonstrates a superior trade-off between accuracy and parameter efficiency when compared to both the baseline and LoRA fine-tuning.

## F IMPLEMENTATION/TRAINING DETAILS

### F.1 IMPLEMENTATION DETAILS

**Base Pre-training.** Building on prior work (Oh et al., 2022), our analysis employs ResNet architectures (ResNet-10 (Guo et al., 2020), ResNet-18, and ResNet-50 (He et al., 2016)) for fair comparison and thorough evaluation. For ResNet-10, we perform supervised pre-training on *mini*ImageNet using Stochastic Gradient Descent (SGD) with a learning rate of 0.1, momentum 0.9, and weight decay $10^{-4}$. For ResNet-18 and ResNet-50, we use ImageNet-1K pre-trained models available in the PyTorch library (Paszke et al., 2019).

**Filter Decomposition.** In our method (Section 3.2), we target only the $3 \times 3$ filters for decomposition, given their dominance, while keeping $7 \times 7$ and $1 \times 1$ filters unchanged. The decomposition, optimized using a least squares objective (5) via coordinate descent (Li & Osher, 2009) over 200 iterations with a sparsity coefficient of $\lambda = 10^{-6}$, is efficiently executed on a GPU (Feinman, 2021), completing within 20 seconds. We decompose the filters into $m \in \{9, 12\}$ spatial filter atoms, where $m = 9$ meets a predefined residual threshold, and $m = 12$ tests overparameterization, suggesting the potential for further decompositions subject to computational resources.

Table 14: 5-way 1-shot and 5-shot accuracy (%) with 95% confidence with different sparsity coefficient ($\lambda$) used at during the *Divide* step, on ResNet-18 and ImageNet-1K, detailing unlabeled data use (first column) and parameters fine-tuned (last column).

| % Unlabeled Data | Method | Sparsity Coefficeint | Dataset | | | | | | | | # of Params. |
|---|---|---|---|---|---|---|---|---|---|---|---|
| | | | ChestX | ISIC | EuroSAT | Crop | CUB | Cars | Places | Plantae | |
| | | | 5-way 1-shot | | | | | | | | |
| 1% | DC-9 | $10^{-4}$ | 22.76±0.42 | 32.99±0.58 | 69.16±0.84 | 83.02±0.82 | 50.50±0.88 | 34.54±0.67 | 60.75±0.86 | 42.13±0.77 | 192.34K |
| | | $10^{-6}$ | 22.87±0.48 | 33.73±0.59 | 69.10±0.85 | 82.35±0.79 | 50.61±0.88 | 34.42±0.67 | 59.50±0.86 | 42.92±0.77 | |
| | | $10^{-8}$ | 22.61±0.43 | 33.26±0.55 | 71.15±0.81 | 83.29±0.82 | 49.49±0.86 | 33.66±0.67 | 60.04±0.85 | 42.68±0.79 | |
| | DC-12 | $10^{-4}$ | 22.72±0.41 | 33.41±0.57 | 69.96±0.84 | 82.91±0.83 | 49.10±0.86 | 33.90±0.65 | 60.04±0.86 | 42.87±0.81 | 192.77K |
| | | $10^{-6}$ | 22.62±0.42 | 34.34±0.59 | 68.88±0.83 | 83.28±0.81 | 49.19±0.86 | 33.61±0.66 | 59.98±0.88 | 43.31±0.81 | |
| | | $10^{-8}$ | 22.71±0.41 | 33.95±0.62 | 69.29±0.84 | 82.59±0.83 | 50.05±0.88 | 33.13±0.63 | 59.34±0.88 | 42.81±0.80 | |
| 5% | DC-9 | $10^{-4}$ | 22.60±0.42 | 35.71±0.63 | 77.23±0.78 | 88.23±0.72 | 49.06±0.87 | 36.58±0.70 | 63.87±0.88 | 46.50±0.84 | 192.34K |
| | | $10^{-6}$ | 22.45±0.42 | 37.20±0.65 | 77.24±0.80 | 88.47±0.70 | 46.34±0.84 | 35.91±0.70 | 64.07±0.86 | 46.29±0.82 | |
| | | $10^{-8}$ | 22.44±0.42 | 36.58±0.61 | 77.34±0.80 | 88.62±0.70 | 48.32±0.86 | 35.89±0.69 | 63.82±0.86 | 46.34±0.85 | |
| | DC-12 | $10^{-4}$ | 22.60±0.42 | 35.71±0.63 | 77.23±0.78 | 88.23±0.72 | 49.06±0.87 | 36.58±0.70 | 63.87±0.88 | 46.50±0.84 | 192.77K |
| | | $10^{-6}$ | 22.48±0.42 | 35.73±0.61 | 77.88±0.76 | 88.60±0.70 | 49.48±0.86 | 36.06±0.70 | 64.27±0.84 | 46.02±0.82 | |
| | | $10^{-8}$ | 22.44±0.42 | 36.58±0.61 | 77.34±0.80 | 88.62±0.70 | 48.32±0.86 | 35.89±0.69 | 63.82±0.88 | 46.34±0.85 | |
| | | | 5-way 5-shot | | | | | | | | |
| 1% | DC-9 | $10^{-4}$ | 26.60±0.45 | 45.56±0.57 | 86.54±0.48 | 94.69±0.39 | 74.08±0.75 | 52.81±0.76 | 80.82±0.58 | 61.14±0.76 | 192.34K |
| | | $10^{-6}$ | 26.63±0.43 | 45.65±0.60 | 86.65±0.48 | 94.42±0.39 | 73.74±0.76 | 52.28±0.75 | 80.09±0.58 | 61.23±0.76 | |
| | | $10^{-8}$ | 26.39±0.45 | 45.56±0.55 | 87.66±0.46 | 94.69±0.40 | 72.63±0.77 | 52.08±0.76 | 80.16±0.58 | 60.94±0.78 | |
| | DC-12 | $10^{-4}$ | 26.38±0.45 | 45.74±0.58 | 87.14±0.47 | 94.52±0.41 | 72.30±0.76 | 51.97±0.76 | 80.53±0.57 | 61.40±0.78 | 192.77K |
| | | $10^{-6}$ | 26.47±0.46 | 46.35±0.57 | 86.34±0.49 | 94.64±0.41 | 71.89±0.78 | 51.25±0.75 | 80.46±0.58 | 61.92±0.75 | |
| | | $10^{-8}$ | 25.85±0.43 | 46.75±0.60 | 87.05±0.48 | 94.46±0.41 | 73.03±0.78 | 50.78±0.76 | 80.07±0.58 | 61.12±0.78 | |
| 5% | DC-9 | $10^{-4}$ | 26.46±0.46 | 48.36±0.62 | 91.64±0.39 | 97.13±0.30 | 70.50±0.77 | 54.77±0.78 | 82.04±0.55 | 64.85±0.79 | 192.34K |
| | | $10^{-6}$ | 26.70±0.45 | 49.96±0.57 | 91.75±0.38 | 97.32±0.29 | 70.25±0.76 | 54.21±0.78 | 81.87±0.55 | 64.63±0.80 | |
| | | $10^{-8}$ | 26.36±0.43 | 50.00±0.59 | 91.66±0.39 | 97.22±0.29 | 69.28±0.78 | 53.52±0.77 | 81.95±0.55 | 64.66±0.79 | |
| | DC-12 | $10^{-4}$ | 26.48±0.44 | 48.56±0.60 | 91.59±0.39 | 97.10±0.30 | 70.17±0.75 | 53.36±0.78 | 81.79±0.56 | 64.61±0.79 | 192.77K |
| | | $10^{-6}$ | 26.67±0.44 | 48.95±0.57 | 91.92±0.38 | 97.37±0.28 | 70.12±0.76 | 53.71±0.79 | 82.10±0.56 | 64.62±0.79 | |
| | | $10^{-8}$ | 26.52±0.44 | 48.56±0.59 | 91.93±0.38 | 97.04±0.30 | 69.80±0.77 | 54.09±0.78 | 81.86±0.56 | 64.33±0.77 | |

Table 15: **5-way 1-shot and 5-shot** accuracies (%) on finetuned **ResNet-18 (ImageNet-1K)** using 1% of target unlabeled samples, our method with different numbers of filter atoms.

| Method | Dataset | | | | | | | | # of Params. |
|---|---|---|---|---|---|---|---|---|---|
| | ChestX | ISIC | EuroSAT | CropDisease | CUB | Cars | Places | Plantae | |
| | 5-way 1-shot | | | | | | | | |
| DC-6 | 22.73±0.44 | 34.20±0.61 | 68.57±0.87 | 82.57±0.81 | 48.56±0.80 | 33.87±0.66 | 59.18±0.87 | 41.86±0.86 | 191.90K |
| DC-9 | 22.87±0.48 | 33.73±0.59 | 69.10±0.85 | 82.35±0.79 | 50.61±0.88 | 34.42±0.67 | 59.50±0.86 | 42.92±0.77 | 192.34K |
| DC-12 | 22.62±0.42 | 34.34±0.59 | 68.88±0.83 | 83.28±0.81 | 49.19±0.86 | 33.61±0.66 | 59.98±0.88 | 43.31±0.81 | 192.77K |
| DC-15 | 22.78±0.42 | 34.20±0.60 | 70.14±0.86 | 82.73±0.82 | 49.68±0.84 | 34.17±0.64 | 59.86±0.88 | 42.26±0.80 | 193.20K |
| | 5-way 5-shot | | | | | | | | |
| DC-6 | 26.44±0.45 | 46.62±0.58 | 85.92±0.53 | 94.44±0.41 | 71.28±0.77 | 51.44±0.75 | 79.21±0.62 | 60.67±0.74 | 191.90K |
| DC-9 | 26.63±0.43 | 45.65±0.60 | 86.65±0.48 | 94.42±0.39 | 73.74±0.76 | 52.28±0.75 | 80.09±0.58 | 61.23±0.76 | 192.34K |
| DC-12 | 26.47±0.46 | 46.35±0.57 | 86.34±0.49 | 94.64±0.41 | 71.89±0.78 | 51.25±0.75 | 80.46±0.58 | 61.92±0.75 | 192.77K |
| DC-15 | 26.39±0.44 | 46.98±0.58 | 87.49±0.46 | 94.39±0.42 | 72.69±0.76 | 52.34±0.74 | 80.52±0.58 | 60.92±0.75 | 193.20K |

**Unsupervised Finetuning.** In our unsupervised fine-tuning, we employ an SGD optimizer with momentum 0.9 and weight decay $10^{-4}$, over 1000 epochs and a batch size of 64. The learning rate starts at 0.1, reducing to 0 via cosine annealing. A two-layer projection head $h$ (Linear-ReLU-Linear) is appended to the extractor $f$, with dimensions of 512 (hidden) and 128 (output). The NT-Xent loss temperature is set at 1.0. For comparisons in the standard setting, we use 20% of $D_T$ as unlabeled data $D_U$ (Phoo & Hariharan, 2021; Oh et al., 2022). Augmentation details are in the supplemental material.

**Few-Shot Evaluation.** During the few-shot evaluation, we attach a linear classifier $g$ to the frozen $f$ as detailed in Section 3.1. We train $g$ using SGD, with settings: batch size 4, learning rate $10^{-2}$, momentum 0.9, and weight decay $10^{-2}$. The evaluation employs labeled target samples $D_L$, distinct from the unlabeled set $D_U$, from $D_T$.

### F.2 BYOL Hyperparameters

For BYOL (Grill et al., 2020), we use the Adam optimizer (Kingma & Ba, 2014), with a set initial learning rate of $3 \times 10^{-5}$, and decayed to 0 via cosine annealing for 1000 epochs. Both the online and target projectors utilize a multilayer perceptron (MLP) architecture comprising two layers, structured as Linear-BatchNorm1D-ReLU-Linear, and featuring a hidden layer dimensionality of 4,096 and a projection layer dimensionality of 256. We maintain the target network's moving aver-

Table 16: 5-way 1-shot and 5-shot accuracies (%) on finetuned ResNet-18 (ImageNet-1K) using 1% of target unlabeled samples, comparing various parameter-efficient methods w.r.t. the number of parameters fine-tuned (last column). Best and second-best results are highlighted in **maroon** and **navy**, respectively.

| Method | Dataset | | | | | | | | # of Params. |
|---|---|---|---|---|---|---|---|---|---|
| | ChestX | ISIC | EuroSAT | CropDisease | CUB | Cars | Places | Plantae | |
| **5-way 1-shot** | | | | | | | | | |
| LoRA-3 (Hu et al., 2021) | 22.38±0.42 | 31.65±0.54 | 69.96±0.89 | **83.36±0.84** | 44.41±0.85 | 32.60±0.65 | 59.07±0.92 | 40.71±0.78 | 218.75K |
| LoHA-1-3 (Yeh et al., 2024) | 22.25±0.40 | 31.28±0.54 | 67.68±0.85 | 81.60±0.83 | 47.51±0.84 | 32.70±0.67 | 59.45±0.91 | 41.37±0.82 | 218.75K |
| LoKrA-32 (Yeh et al., 2024) | 22.68±0.44 | 32.11±0.61 | **70.57±0.84** | 82.32±0.81 | 48.72±0.84 | 33.69±0.68 | 59.23±0.88 | 42.34±0.80 | 213.95K |
| **DC-6 (Ours)** | 22.73±0.44 | 34.20±0.61 | 68.57±0.87 | 82.57±0.81 | 48.56±0.80 | 33.87±0.66 | 59.18±0.87 | 41.86±0.86 | **191.90K** |
| **DC-9 (Ours)** | **22.87±0.48** | 33.73±0.59 | 69.10±0.85 | 82.35±0.79 | **50.61±0.88** | **34.42±0.67** | 59.50±0.86 | 42.92±0.77 | **192.34K** |
| **DC-12 (Ours)** | 22.62±0.42 | **34.34±0.59** | 68.88±0.83 | **83.28±0.81** | 49.19±0.86 | 33.61±0.66 | **59.98±0.88** | **43.31±0.81** | **192.77K** |
| **DC-15 (Ours)** | 22.78±0.42 | 34.20±0.60 | 70.14±0.86 | 82.73±0.82 | 49.68±0.84 | **34.17±0.64** | 59.86±0.88 | 42.26±0.80 | **193.20K** |
| **5-way 5-shot** | | | | | | | | | |
| LoRA-3 (Hu et al., 2021) | 24.79±0.43 | 41.93±0.57 | 84.76±0.54 | 93.33±0.46 | 63.13±0.81 | 47.46±0.75 | 75.92±0.64 | 57.27±0.77 | 218.75K |
| LoHA-1-3 (Yeh et al., 2024) | 25.28±0.43 | 44.26±0.54 | 86.32±0.52 | 93.48±0.45 | 69.16±0.81 | 47.26±0.73 | 77.01±0.62 | 58.05±0.78 | 218.75K |
| LoKrA-32 (Yeh et al., 2024) | 26.45±0.44 | 43.44±0.57 | 86.32±0.47 | **94.52±0.39** | 72.50±0.76 | 49.32±0.75 | 79.85±0.57 | 61.00±0.78 | 213.95K |
| **DC-6 (Ours)** | 26.44±0.45 | 46.62±0.58 | 85.92±0.53 | 94.44±0.41 | 71.28±0.77 | 51.44±0.75 | 79.21±0.62 | 60.67±0.74 | **191.90K** |
| **DC-9 (Ours)** | **26.63±0.43** | 45.65±0.60 | **86.65±0.48** | 94.42±0.39 | **73.74±0.76** | **52.28±0.75** | 80.09±0.58 | **61.23±0.76** | **192.34K** |
| **DC-12 (Ours)** | 26.47±0.46 | 46.35±0.57 | 86.34±0.49 | **94.64±0.41** | 71.89±0.78 | 51.25±0.75 | 80.46±0.58 | 61.92±0.75 | **192.77K** |
| **DC-15 (Ours)** | 26.39±0.44 | **46.98±0.58** | **87.49±0.46** | 94.39±0.42 | 72.69±0.76 | 52.34±0.74 | 80.52±0.58 | 60.92±0.75 | **193.20K** |

Table 17: 5-way 1-shot and 5-shot accuracies (%) on a ConvNext-Tiny (ImageNet-1K) (Liu et al., 2022) finetuned using 1% of unlabeled target samples. Best and second-best results are highlighted in **maroon** and **navy**, respectively.

| Method | Dataset | | | | | | | | # of Params. |
|---|---|---|---|---|---|---|---|---|---|
| | ChestX | ISIC | EuroSAT | CropDisease | CUB | Cars | Places | Plantae | |
| **5-way 1-shot** | | | | | | | | | |
| SimCLR (Chen et al., 2020) | **21.71±0.40** | 23.08±0.48 | 43.18±0.11 | 52.90±0.93 | **28.85±0.62** | **24.53±0.58** | 28.00±0.013 | 26.82±0.57 | 27820.13K |
| LoRA-3 (Hu et al., 2021) | 20.56±0.36 | 22.53±0.45 | 39.82±0.86 | 44.34±0.88 | 27.18±0.58 | 23.09±0.47 | 29.81±0.63 | 24.82±0.53 | 1838.26K |
| **DC-9 (Ours)** | **21.62±0.40** | 24.83±0.51 | 42.03±0.38 | **55.94±0.92** | 27.31±0.60 | 24.01±0.51 | 26.45±0.53 | 25.90±0.57 | 1823.68K |
| **DC-12 (Ours)** | 21.55±0.38 | **24.90±0.54** | 45.42±0.91 | 51.84±0.90 | 28.25±0.60 | **24.77±0.53** | 33.79±0.65 | **27.56±0.59** | 1826.33K |
| **5-way 5-shot** | | | | | | | | | |
| SimCLR (Chen et al., 2020) | 23.67±0.40 | 29.62±0.48 | **56.43±0.27** | **73.26±0.77** | **35.04±0.59** | 27.24±0.62 | 37.02±0.58 | **34.62±0.59** | 27820.13K |
| LoRA-3 (Hu et al., 2021) | 21.54±0.35 | 26.02±0.47 | 50.48±0.76 | 59.46±0.84 | 31.56±0.55 | 26.82±0.50 | **38.86±0.64** | 30.64±0.54 | 1838.26K |
| **DC-9 (Ours)** | **24.00±0.42** | 32.04±0.53 | 55.98±0.50 | **77.25±0.71** | 33.58±0.59 | **28.84±0.53** | 33.58±0.55 | 33.38±0.59 | 1823.68K |
| **DC-12 (Ours)** | 23.82±0.42 | **33.71±0.52** | 60.73±0.74 | 72.87±0.75 | 34.48±0.58 | 29.79±0.55 | 45.04±0.66 | 35.03±0.61 | 1826.33K |

age coefficient at 0.99. Subsequent to the online projector, the predictor also adopts a two-layer MLP design, with a hidden layer dimensionality of 4,096 and an output dimensionality for prediction set at 256.

## F.3 Augmentations used for SimCLR and BYOL

- `RandomResizedCrop`: Randomly crop a portion of an image and then resize it to $224 \times 224$.

- `RandomColorJitter`: Randomly change the brightness, contrast, and saturation, with a probability of 1.0.

- `RandomHorizontalFlip`: Randomly flip an image on a vertical axis, with a probability of 0.5.

- `RandomGrayscale`: Randomly convert image into grayscale, with a probability of 0.1.

- `RandomGaussianBlur`: Randomly blur an image with Gaussian blur of kernel size $5 \times 5$, with a probability of 0.3.

