# OpenReview forum: "Divide and Conform: Unleashing Spatial Filter Atoms for Unsupervised Target Transferability"
_ICLR.cc/2025/Conference — ICLR 2025 Conference Withdrawn Submission_

### Official Review · Reviewer_mEe3 · 2024-11-02

**Soundness:** 2
**Presentation:** 2
**Contribution:** 2
**Rating:** 5
**Confidence:** 4

**Summary:**

This paper argues that directly fine-tuning pre-trained models carries the risk of insufficiently leveraging the foundational knowledge accumulated during pre-training, which may adversely affect performance on target tasks. To address this issue, this paper decomposes the convolution kernel into two components: spatial filter atoms and atom-coefficients. During the fine-tuning phase, this paper only fine-tunes the spatial filter atoms, thereby achieving fine-tuning with fewer parameters.

**Strengths:**

+：In the context of parameter-efficient tuning, this paper decomposes convolutional kernels into two components, and focuses on fine-tuning spatial filter atoms while retaining existing knowledge to effectively transfer to target tasks. This approach provides a different perspective on fine-tuning tasks in convolutional layers

+: The method is straightforward and easy to understand.

**Weaknesses:**

-: The motivation of this paper is not very clear. As stated in line 66-71, this paper points out three challenges: (1) lack of the base dataset, (2) heavy computational cost of full finetuning, and (3) scarcity of labeled target data. They are common issues and have been well studied in previous parameter-efficient tuning works. Therefore, what are advantages of this work over previous parameter-efficient tuning works?

-: The technical contribution of this work could be further clarified. Particularly, compared to previous parameter-efficient tuning works, this paper relies on convolutional kernel decomposition techniques. Therefore, the authors would better discuss the advantages of the introduced convolutional kernel decomposition over existing works.

Additionally, dictionary learning for kernel decomposition involved in this work is complex and will bring significant computational overhead. More importantly, how about its scalability to large-scale convolutional neural works and vision transformers. Particularly, vision transformers are most widely used as backbones of pre-trained models.

For the experiment part, it seems to lack sufficient analysis on why only fine-tuning of the spatial filter atoms yields effective results. Furthermore, it also lacks experimental support on how freezing of the atom-coefficients preserves the knowledge of the pre-trained network. Additionally, the authors would better conduct more experiments with larger convolutional kernels and the larger model sizes to show the generalizability of the proposed methods.

**Questions:**

How is LoDC implemented on ResNet18?

---

### Official Review · Reviewer_kZRN · 2024-11-03

**Soundness:** 4
**Presentation:** 4
**Contribution:** 3
**Rating:** 5
**Confidence:** 2

**Summary:**

Authors introduce Divide and Conform, aimed at augmenting the transferability of pre-trained
convolutional neural networks (ConvNets), in the absence of base data.
It is a two step process , spatial only convolutions and channel combination.

Authors claim that their approach is designed to enhance the adaptability of pre-trained models to specific
target tasks, while assuming only a limited amount of unlabeled data is available for the target task and no access
to the extensive base dataset, achieving all this in a parameter-efficient manner.

**Strengths:**

The paper is well written and easy to follow.
Detailed anaylysis has been done on several datasets.
Several quantitative results are presented.

**Weaknesses:**

I take from the results on EuroSAT dataset message that the proposed method found it hard to learn discriminative features
as compared to other methods.
I would be great to see some qualitative results.
I am new to this direction of research, for me, I am trying to see sparsity and your method together, how are they different or similar_

**Questions:**

Please follow weaknesses.

---

### Official Review · Reviewer_3kt8 · 2024-11-04

**Soundness:** 2
**Presentation:** 2
**Contribution:** 2
**Rating:** 5
**Confidence:** 4

**Summary:**

Exploring the transfer of pre-trained model knowledge in cross-domain few-shot learning settings is valuable. This manuscript introduces "Divide and Conform," a method designed to enhance the transferability of pre-trained convolutional neural networks (ConvNets) without relying on base data. The approach involves fine-tuning only the decomposed spatial filter atoms while keeping the atom-coefficients frozen, facilitating cross-domain transfer. Evaluated on multiple benchmark datasets, the proposed method demonstrates efficient knowledge transfer with minimal parameter adjustment.

**Strengths:**

1. Exploring new efficient fine-tuning methods for transferring diverse pre-trained models is meaningful.

2. The paper provides a clear and logical description and definition of the proposed method.

3. The approach achieves comparable results.

**Weaknesses:**

1. The manuscript elaborates in detail on research progress in cross-domain few-shot learning and spatial filtering decomposition in the second section (Related Works). However, discussions of the latest related works are lacking. Additionally, the connections and distinctions between this manuscript and existing works in the field should be carefully explained.

2. The arrangement of tables and figures should align with the textual content to facilitate reader comprehension and comparison.

3. In the experimental section, the manuscript should include comparisons with the latest cross-domain few-shot learning methods. Additionally, SimCLR is a straightforward framework for contrastive learning of visual representations. The authors should compare their approach with more mainstream and efficient parameter-tuning methods, such as vision prompt tuning.

4. The experimental results indicate that the proposed method does not outperform all baselines comprehensively. The authors should provide a detailed explanation of this.

**Questions:**

1.In this manuscript, the authors conduct a detailed comparison of the proposed method’s performance with SimCLR and LORA-style methods under a cross-domain few-shot learning setting. Would the task benefit from stronger parameter-tuning performance under the setups of SimCLR and LORA-style methods?

2.Does the proposed method still demonstrate an advantage on more challenging benchmarks?

---

### Note · Authors · 2024-11-13

I have read and agree with the venue's withdrawal policy on behalf of myself and my co-authors.